# Emergence and Effectiveness of Task Vectors in In-Context Learning: An Encoder Decoder Perspective

Seungwook Han [* 1]   Jinyeop Song [* 1]   Jeff Gore [1]   Pulkit Agrawal [1]

## Abstract

Autoregressive transformers exhibit adaptive learning through in-context learning (ICL), which begs the question of how. Prior works have shown that transformers represent the ICL tasks as vectors in their representations. In this paper, we leverage the encoding-decoding framework to study how transformers form task vectors during pretraining and how their task encoding quality predicts ICL task performance. On synthetic ICL tasks, we analyze the training dynamics of a small transformer and report the coupled emergence of task encoding and decoding. As the model learns to encode different latent tasks (e.g., "Finding the first noun in a sentence.") into distinct, separable representations, it concurrently builds conditional decoding algorithms and improves its ICL performance. We validate this phenomenon across pretrained models of varying scales (Gemma-2 2B/9B/27B, Llama-3.1 8B/70B) and over the course of pretraining in OLMo-7B. Further, we demonstrate that the quality of task encoding inferred from representations predicts ICL performance, and that, surprisingly, finetuning the earlier layers can improve the task encoding and performance more than finetuning the latter layers. Our empirical insights shed light into better understanding the success and failure modes of large language models via their representations.

## 1. Introduction

Throughout history, humans have made sense of the world by distilling complex experiences into fundamental abstractions, such as physics and mathematics. These mental models enable us to learn quickly, predict outcomes, and adapt to new situations. In artificial intelligence, autoregressive transformers are beginning to exhibit similar capabilities (Brown et al., 2020; Bubeck et al., 2023; Ajay et al., 2023; Han et al., 2024). Through in-context learning (ICL), they adapt to new tasks without parameter updates, suggesting they might also be forming internal abstractions (Raventós et al., 2024; Hong et al., 2024; Zheng et al., 2024; Kumar et al., 2024; Han et al., 2024).

Hendel et al. (2023); Merullo et al. (2023); Todd et al. (2023) introduce a mechanistic perspective on how pretrained LLMs represent the latent concepts underlying the ICL task as vectors in their representations. They empirically demonstrate that these task-specific vectors can trigger the desired ICL behavior in many cases, with the effectiveness varying across tasks. Although an impactful first observation, there still remains unanswered questions of why these task vectors exist in the first place and why the effectiveness varies by task. This necessitates a deeper mechanistic understanding of this internal abstraction behavior of LLMs.

In our work, we leverage the encoding-decoding framework to investigate the origin and formation of task vectors in transformers. To study their emergence during pretraining, we train a small transformer on a mixture of sparse linear regression tasks. We find that **task encoding** emerges as the model learns to map different latent tasks into *distinct*, *separable representation spaces*. This geometric structuring of the representation space is coupled with the development of task-specific ICL algorithms – namely, **task decoding**. Through causal analysis, we demonstrate that the model associates different algorithms to different learned concepts in the representation space and that ICL happens through the two-step process. Importantly, we see that the **emergence of the two-stage process coincides with each other**, suggesting a mutual dependence between the two.

Inspired by these findings, we investigate the task encoding-decoding phenomenon across different pretrained model families and scales (Llama-3.1-8B/70B and Gemma-2 2B/9B/27B) on more natural ICL tasks, such as part-of-speech tagging and bitwise arithmetic. We introduce task decodability, a geometric measure that quantifies how effectively a model can decode the task from its intermediate

*Equal contribution [1]Massachusetts Institute of Technology, Cambridge, USA. Correspondence to: Seungwook Han <swhan@mit.edu>, Jinyeop Song <yeopjin@mit.edu>.

*Proceedings of the 42nd International Conference on Machine Learning*, Vancouver, Canada. PMLR 267, 2025. Copyright 2025 by the author(s).

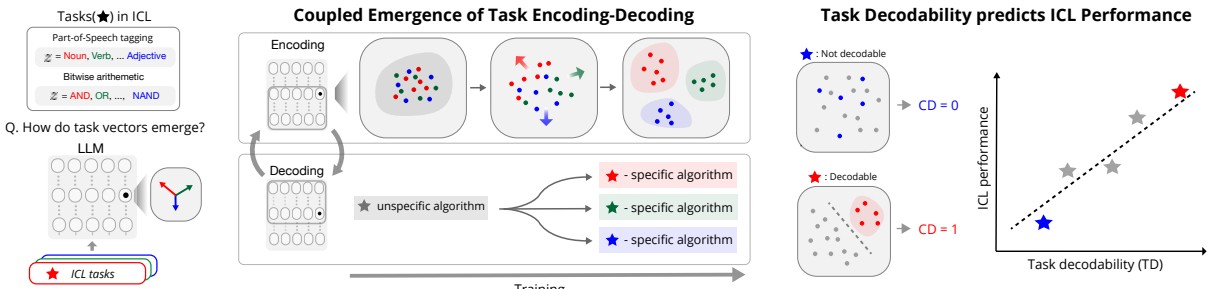

*Figure 1.* **An overview of our work.** We study the **task encoding-decoding** to explain why and how task vectors emerge in pretrained LLMs. We demonstrate that transformers concurrently learn to map latent concepts into separable representations and develop task-specific decoding algorithms. We validate the generality of this finding across model families and scales, and show that the quality of task encoding-decoding can predict ICL task performance.

representations. We demonstrate that task decodability is causally related to ICL performance through ablation studies on ICL examples, finetuning, and prompting. Overall, our study reveals that the encoder-decoder frameworks provide valuable insights into the emergence of task vectors and why their effectiveness varies across tasks.

Our main contributions are as follows:

1. We first study the emergence of task vectors during training under the encoder-decoder framework. By training a small transformer on synthetic ICL tasks (§3.3), we observe the coupled occurrence of task encoding and task decoding, ultimately forming task vectors.

2. We introduce Task Decodability (TD) as a geometric measure to quantify how well the model can infer the latent tasks from its representations and demonstrate that TD effectively predicts downstream ICL performance in pretrained LLMs (§4.2). We demonstrate our framework's generality across tasks, model families, and scales (Llama 3.1 8B/70B, Gemma 2B/9B/27B) and further study its evolution throughout pretraining using different checkpoints of OLMo-7B.

3. We establish the causal relationship between TD and ICL performance in pretrained LLMs through mechanistic intervention (§4.1) and controlled finetuning (§4.4). Contrary to common convention, we show that finetuning the earlier layers improves ICL performance more than finetuning the later layers (Wu et al., 2024; Kumar et al., 2022).

4. We offer an unifying perspective on how the learning signal of more in-context examples, finetuning, and prompting (§4.5) materializes in LLMs.

## 2. Related Work

**Mechanisms of ICL.** Astounded by LLMs' ability to perform ICL, many have proposed theories to understand the mechanisms of ICL. Some works (Dai et al., 2023; von Oswald et al., 2023; Ahn et al., 2024; Akyürek et al., 2024) have proposed that LLMs, with linear attention (Katharopoulos et al., 2020), can implement stochastic gradient descent to perform ICL. Other works (Xie et al., 2021; Wang et al., 2024; Ye et al., 2024) have presented a Bayesian framework to theoretically explain the workings of ICL. This view implies that the model implements a two-stage algorithm to estimate the posterior $P(z|\mathcal{D})$ and the likelihood $P(y_*|x_*, \mathcal{D})$. In this work, we adopt this framework and demonstrate how the model implements it through its intermediate representations. More specifically, we study the emergence of task encoding and decoding.

**Task Vectors and Latent Concepts in LLM Representations.** Recent work by Todd et al. (2023) and Hendel et al. (2023) identifies task-specific vectors in LLMs that can induce desired in-context learning behaviors (e.g., object-color mapping). Building on this foundation, our study examines when and how task-specific representations emerge and how their quality (TD score) can be measured and used to predict downstream ICL performance. Moreover, Park et al. (2024) study the different algorithmic stages for ICL under homogeneous Markov chain settings. In contrast, our work expands the study to heterogeneous tasks, a regime more reflective of LLM pretraining, and proposes a more general encoding-decoding framework to understand the mechanics of ICL.

Beyond the scope of task-specific vectors, several studies have explored how language models encode a wide range of latent concepts (Dalvi et al., 2022; Merullo et al., 2023), including truthfulness (Marks & Tegmark, 2023), time, and space (Gurnee & Tegmark, 2024). These investigations reveal that such notions can be linearly separable in the hidden

representations, and that model scaling often yields more interpretable features (Bricken et al., 2023; Cunningham et al., 2023)

**Mechanistic Interpretability.** To study the causal relationship between the quality of task encoding-decoding and downstream ICL performance, we adopt causal mediation analysis techniques from Geiger et al. (2020); Vig et al. (2020); Todd et al. (2023); Heimersheim & Nanda (2024); Merullo et al. (2024). We specifically use the method of activation patching, where we replace the activations of an intermediate layer from a sample with another.

## 3. Understanding In-context Learning

### 3.1. Notation and Background

We focus on ICL problems, where the goal is to predict $y_*$ from a query $x_*$, given some in-context examples $\mathcal{D} = \{(x_i, y_i)\}_{i=1}^n$. Each problem shares a latent task $z$ that links inputs $x$ to outputs $y$. For instance, in an ICL task where the latent task is object-color mapping, we provide demonstrations like (apple, red), (banana, yellow), and (grape, purple), and then ask for what comes after (lemon, ?). We employ this parameterization to accommodate latent tasks varying in complexity, from simple function regression problems (Garg et al., 2022; von Oswald et al., 2023; Li et al., 2023) to Part-of-Speech (POS) tagging (Blevins et al., 2022; Banko & Moore, 2004) and bitwise arithmetic (He et al., 2024).

### 3.2. Theoretical Framework

Of the many different frameworks (Bai et al., 2024; Min et al., 2022; von Oswald et al., 2023; Akyürek et al., 2024) to understand the workings of ICL, we adopt the Bayesian view (Xie et al., 2021; Mittal et al., 2024; Wang et al., 2024; Ye et al., 2024). It proposes that transformers implicitly infer the latent variable $z$ underlying the demonstrations and apply it to generate an answer. More formally,

$$p(y_* \mid x_*, \mathcal{D}) = \int_{\mathcal{Z}} P_\theta(y_* \mid x_*, z)\, P_\theta(z \mid \mathcal{D})\, dz \quad (1)$$

This framework suggests ICL is a two stage process. First, *latent concept inference*. Latent concept $z$ is approximated from $\mathcal{D}$ through the distribution $\hat{z} \sim P_\theta(z|\mathcal{D})$. Second, *selective algorithm application*. The model applies an algorithm conditioned on $\hat{z}$ to predict $y_*$ as given by $P_\theta(y_*|x_*, \hat{z})$.

Although theoretically compelling, it was not until recently that Hendel et al. (2023); Todd et al. (2023); Merullo et al. (2023) showed empirical evidence of models encoding the latent concepts in the intermediate representations. They illustrate that task-specific vectors are then decoded and trigger the desired ICL task behavior. With a simple encoder-decoder analogy, these findings suggest that the two-stage behavior of ICL, as described in Equation (1), is mediated by the encoding and decoding of latent variables within the representation space. Building on this idea, we begin our investigation with the following questions:

1. How do task vectors emerge during training, and what drives their varying effectiveness across tasks?

2. How is the model's ability to accurately infer the latent tasks related to downstream ICL performance?

### 3.3. Motivation: Synthetic Experiments

As a motivating experiment, we study the formation of task vectors during training dynamics of a small autoregressive transformer on synthetic ICL tasks. We observe that, as the model "localizes" the latent task by building a distinct representation from the others, it associates it with a uniquely corresponding decoding algorithm. Building on this observation, we outline the task encoding and decoding framework to explain the formation and mechanism of task vectors.

**Task.** We compose our task as a mixture of sparse linear regression. We follow the conventional linear regression setup from Garg et al. (2022); von Oswald et al. (2023) and construct the input-output pair $(x_i, y_i)$ by sampling $x_i \sim \mathcal{N}(0, \boldsymbol{I_D})$ and $y_i = W^T x_i + \epsilon_i$, where $W$ is randomly generated from a standard normal distribution, $\mathcal{N}(0, \boldsymbol{I_D})$, and $\epsilon_i \sim \mathcal{N}(0, \sigma^2)$. We, however, add sparsity constraints to $W$ with the sparsity pattern represented by the basis $B_k$. Each $B_k$ has a rank of $r$. In other words, the basis chooses the dimensions of $W$ to turn on and off. The basis is sampled uniformly from $\mathcal{B} = \{B_1, B_2, B_3, B_4\}$ and each basis is non-overlapping and orthogonal to each other. By default, we set $D = 16$ and $r = 4$. By adding this layer of latent concept of $\boldsymbol{B}$, we can explicitly control and interpret the latent concepts, and analyze their representations.

**Model and Training.** We train a 12-layer GPT-2 architecture transformer (Radford et al., 2019) with an embedding dimension of 128 and 8 attention heads. We train the model to minimize mean squared error (MSE) loss over the sequence length of 20. We run 5 different random seeds for training and report observations that generalize across the runs. We detail the experimental setup in Appendix D.2.

**Theoretical Error Bounds.** The error bounds of regression depend on whether the model learns to infer the underlying bases. If the model can infer the bases, then the model can theoretically achieve $r$-dimensional

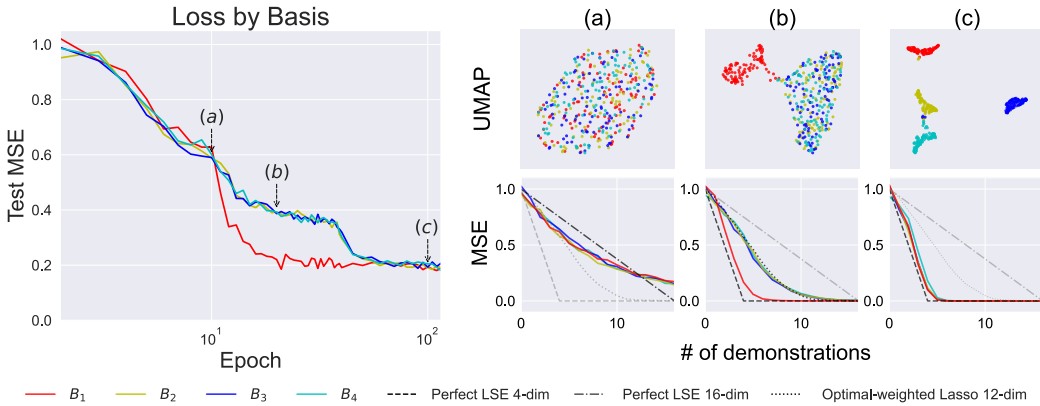

*Figure 2.* **Coupled emergence of task encoding and conditional decoding algorithms** in mixture of sparse linear regression. The loss curve on the left-hand side shows different convergence dynamics per basis and show three phases of descent, which we mark with (a), (b), and (c). The test MSE is the mean squared error computed over a sequence of 20 in-context examples. On the right-hand side, we plot the geometric changes in the representations and how they separate by basis at these marked points. These points coincide with the algorithmic switching behavior. For the UMAP visualizations, we randomly draw 100 samples for each basis. Next, we collected $y$-token representations of 5th layer at 20th demonstrations and plotted the UMAP with the parameters of n neighbors=15 and min dist=0.1.

regression, where the MSE approaches 0 with $r$ in-context examples. If not, the model, in the worst case, can perform $D$-dimensional regression with $r$-sparsity, which has a longer tailed error curve that approaches 0 with $D$ in-context examples. With these insights, we can better analyze which latent basis the model has learned and the associated algorithm from its error curve. Note that we define "algorithm" as a class of statistical methods for linear regression, as detailed in Appendix D.1.

**Observation 1: Different Loss Dynamics per Basis.** We interestingly observe that each basis, despite having identical task complexities, exhibits different loss descents during training. Figure 2 shows the test MSE averaged over the sequence over training. $B_1$ displays a distinct loss descent dynamic, undergoing an abrupt drop at epoch 10. In contrast, the other three bases, $B_2$, $B_3$, and $B_4$, exhibit correlated loss descent dynamics, with two smaller descents at 10 and 40 epochs. This suggests that the model learns to infer $B_1$ differently and applies a selective decoding algorithm.

**Observation 2: Emergence of Separable Representations and Coupled Algorithmic Phase Transitions.** We also analyze the geometry of the intermediate representations at layer 5 to question how the model may be encoding the latent bases. Surprisingly, at the three points of descent (a, b, c) marked in Figure 2, the model gradually builds separate representations for the different bases as shown in the UMAP visualizations. At point (a), the three bases are clustered together and the model's algorithm resembles a 16-dimensional weighted LASSO regression. As $B_1$ separates out at point (b), the model starts to leverage the inferred basis to switch to a 4-dimensional regression. At point (c), when all four classes are separable, the model converges

to the optimal 4-dimensional regression. This observation suggests that model encodes the tasks into separated representations to conditionally apply decoding algorithms.

**Causal Relation between Task Encoding and Performance.** We conduct perturbation analysis to validate that the model conditionally applies decoding algorithms based on the separated representations. Given an input of a source basis, we patch the activations of layer 5 – representations of the residual stream of the transformer layer – with the mean activations of a target basis and analyze whether it will improve or degrade performance. We specify layer 5 for this activation patching analysis because the separation of representations by concept is only clearly observed from that layer and afterwards based on the UMAP visualizations in Figure 11. We formally describe the activation patching procedure in Appendix C. When the source is equal to the target (*self-perturbation*), the patching should help the model identify the basis and improve performance. Otherwise, it should hinder correct basis inference and therefore degrade performance. We perform this analysis at different steps of training – (b) and (c) from Figure 2, when the latent task representations are semi and fully separable.

In Figure 7 of Appendix D.3, we present the perturbation analysis at point (b) on the left. In this case, $B_{2,3,4}$ forms one cluster and $B_1$ another. We observe that all the self-perturbations along the diagonal and intracluster ($B_{2,3,4}$) slightly decrease the loss or show no effect. However, when we apply perturbations across different clusters, the loss spikes, indicating that we trigger different decoding algorithms unsuitable for the input sequence. This analysis shows that, because the model was only able to encode two different latent concepts in the intermediate representations,

it only learns two classes of algorithms, one for $B_1$ and another for $B_{2,3,4}$.

On the right of Figure 7, we conduct the same study at convergence, when the model learns to encode all of the latent concepts as distinct representations. Surprisingly, we observe that the model undergoes an algorithmic phase transition of implementing individual task-specific algorithms. Not only does all the self-perturbation along the diagonal improve performance more noticeably, but also any perturbation to a different basis results in significantly higher losses.

These results altogether draw the picture that a transformer, when trained to perform ICL, gradually learns to encode the latent tasks into separable representation spaces and learns to conditionally apply decoding algorithms *simultaneously*. These observations suggest that task encoding and decoding are mutually dependent.

**Generalizability to Increasing Dimensions and Basis Overlap** We investigate this coupled emergence of task encoding and decoding under more complex settings. To this end, we increase the number of tasks and introduce non-orthogonal bases with overlaps (i.e., some bases are correlated). We describe the specific details in Appendix D.4. First, we, once again, observe the coupled emergence of task encoding and decoding in these more complex tasks in Figure 13 of Appendix D.4. Second, as shown in Figure 15 of Appendix D.4, a more intriguing pattern emerges: bases that share overlap and are correlated, even at convergence, are not fully separated in the representation space and share the same loss over training. This suggests that, in natural text, where tasks are often correlated and exhibit semantic overlaps, the model may similarly fail to disentangle their representations and distinguish these tasks.

**Generalizability to Mixture of Various Regression Tasks** To further validate the robustness of the task encoding-decoding framework, we trained a transformer using a mixture of various regression tasks (linear, sparse linear, leaky ReLU, and quadratic regression) from Li et al. (2023) detailed in Appendix F. In Figure 16, the model successfully learns linear, sparse linear, and leaky ReLU regression but fails at quadratic regression. Despite strong ICL performance, intermediate representations across the regression tasks in intermediate layers have high overlap. We hypothesize that task vectors do not naturally emerge here because these tasks could be solved with the same core linear regression algorithm (Li et al., 2023).

To verify the conjecture, we perform attention head pruning experiments detailed in Appendix F and report that the different regression families, excluding quadratic regression, share the same decoding algorithm. We sequentially prune

each attention head and measure the resulting change in MSE. Figure 17 shows that in the mixture of regression tasks, there is a consistent sharing of attention heads across linear regression, leaky ReLU, and sparse linear regression tasks, excluding quadratic regression. This contrasts with the head pruning result of the sparse linear regression in Figure 18, where each basis seemingly operates with a different set of attention heads and algorithm. This suggests structural similarity in the algorithms in this new mixture setting and provides a direct evidence that they are structurally indistinguishable within the model, supporting our hypothesis.

### 3.4. Task Encoding-Decoding

Based on the observations above, we define *task encoding and decoding* that serves as the core framework throughout the paper. The formal definition is in Appendix B.

> **Definition 3.1** (Task Encoding and Decoding). Over training, transformers learn separable representations by latent tasks – **task encoding**. Simultaneously, the model learns task-specific algorithms by leveraging the separable representation spaces – **task decoding**. We illustrate that these two processes combined manifest as task vectors, and that they emerge concurrently during training.

## 4. Towards Natural Experiments

In this section, we empirically study the task encoding-decoding phenomenon in pretrained LLMs. Specifically, we test several hypotheses driven by the the proposed task encoding-decoding framework and demonstrate that our geometric measure - called Task Decodability - can serve as a proxy of the quality of task vector and ICL performance.

**Tasks.** We construct two classes of algorithmic tasks – natural language processing and arithmetic – comprising a total of 12 tasks. Within each class, the tasks are designed to be semantically similar, ensuring that the input distributions are alike across tasks. While the underlying tasks differ (e.g., different arithmetic operations or linguistic patterns), the surface features of inputs remain consistent. By keeping the input distributions similar, we can effectively assess the model's ability to infer and encode latent concepts based solely on subtle differences in the data, rather than simply relying on the input variations. Refer to Appendix G for more details.

*Part-of-Speech (POS) tagging.* We construct a POS tagging (Blevins et al., 2022; Banko & Moore, 2004) dataset from Marcus et al. (1994), consisting of POS tags, such as Noun, Adjective, Verb, Adverb, Preposition, and Pronoun. Given

an input text and hidden POS tag $z_i$ (e.g., Noun), one needs to output the first word that is of the specified POS tag.

*Bitwise arithmetic.* We construct a bitwise arithmetic dataset consisting of 6 different operators, AND, NAND, OR, NOR, XOR, and XNOR. Given a pair of 5-digit binary numbers and the hidden operator $z_i$ (e.g., AND), one needs to output the resulting binary number after the operation.

For both of these tasks, we create an additional Null class, for which there is no latent concept. In bitwise arithmetic, the Null operator outputs random binary digits, and in POS tagging, the Null class pairs the input sentences with a randomly selected word. This task helps us identify the cases in which the model is confused about the concept.

**Model.** We use Llama-3.1-8B as the default model for the experiments. We test our hypothesis across different families and scales (Gemma-2 2B/9B/27B and Llama-3.1-8B/70B) in Section 4.1 and Appendix G. For analyzing the models over the course of pretraining, we use the checkpoints of OLMo 7B (Groeneveld et al., 2024). We do not train any model, except when we study the causal effect of task decodability by finetuning in Section 4.4. We further detail the experimental setup in Appendix G.

**Evaluation.** We evaluate the performance of the model on different tasks by computing the exact-match accuracy between the generated output under greedy decoding and the ground truth. All of the evaluations assume 4-shots of examples, unless specified otherwise.

**Task Decodability (TD).** To quantify how the task encoding progresses, we employ a simple $k$-Nearest Neighbor (k-NN) classification metric. Inspired by prior studies using linear probes (Rimanic et al., 2020; Alain & Bengio, 2018), we assess whether the latent task can be extracted in a simple manner from their representations. Specifically, we use the representations of the token immediately before $y_*$ at a chosen layer and predict the latent task by majority voting among its $k$ nearest neighbors ($k = 10$, $N = 100$). For choosing which layer to measure TD, we compute TD across all layers and choose the layer that best encodes the task. We empirically validate with UMAP visualizations that the representations become separable precisely when the TD score peaks (See Figures 10, 11 and 19).

We now formally define TD. Let $\mathcal{T}$ be the set of latent tasks. For each task $z \in \mathcal{T}$, sample $N = 100$ datapoints $\{(x_i, y_i)\}_{i=1}^N$ and collect the intermediate representations $\{h_i\}_{i=1}^N$ from a chosen layer (e.g., the token embedding immediately after $x_i$). Label each representation $h_i$ with the corresponding task $z$. This yields a set $\mathcal{S} = \bigcup_{z \in \mathcal{T}} \{(h_i, z) \mid i = 1, \ldots, N\}$, over all tasks $z$.

Given a query point $(h_i, z)$, we exclude $(h_i, z)$ from $\mathcal{S}$ (i.e.,

$\mathcal{S} \setminus (h_i, z)$) and find its $k$ nearest neighbors (with $k = 10$) in the remaining set. We then use majority voting on these neighbors' task labels to produce a predicted label $\hat{z}$. If $\hat{z} = z$, classification for the task label is correct. The TD score at this layer is the fraction of query points classified correctly. TD $= \frac{1}{|\mathcal{D}|} \sum_{(h_i, z) \in \mathcal{D}} \mathbf{1}[\hat{z} = z]$.

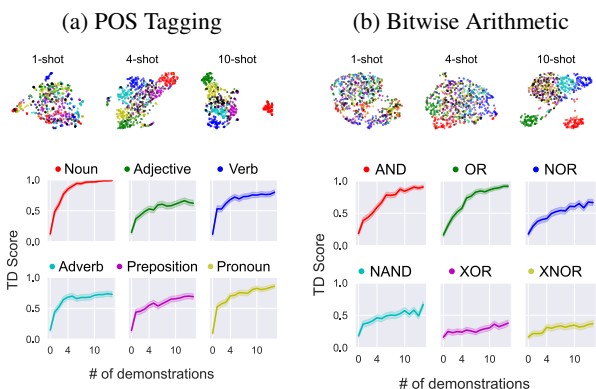

*Figure 3.* **Task encoding in Llama-3.1-8B.** (Top) UMAP of the intermediate representations and (Bottom) TD scores for each tasks at layers 15 and 13, respectively, for POS tagging and bitwise arithmetic with varying numbers of in-context examples.

### 4.1. Task Encoding and Decoding in Pretrained LLMs

> **Hypothesis 1**: In pretrained LLMs, task decodability varies across tasks and determines the effectiveness of the task vector.

**Task decodability varies across tasks.** We begin by qualitatively analyzing the intermediate representations of LLMs across different tasks using UMAP. As shown in Figure 3, the degree of localization of representations depends on the task. For instance, tasks like AND, OR, Noun, and Pronoun exhibit clear and distinct clusters when sufficient in-context examples (e.g., 10-shots) are provided, while tasks like XNOR and XOR in bitwise arithmetic or Adjective and Preposition in POS tagging remain overlapped with the Null class. This suggests that the model inherently learns certain tasks better than others during pretraining, likely due to differences in their predictability or frequency in the pretraining corpus. Importantly, the emergence of separability with increasing examples highlights that in-context examples act as a signal for materializing task-specific representations.

To better quantify such variability of representational separations between different tasks in Figure 3, we use our proposed Task Decodability (TD) scores. Analogous to the UMAP visualizations, we observe that some tasks, such as Noun and Pronoun in POS tagging and AND and OR in bitwise arithmetic, are much more decodable from their rep-

resentations than the others. Also, we confirm in Figure 19a of Appendix G.1, TD score peaks in the middle, suggesting earlier layers encode the task and latter layers execute the decoding algorithm.

**Task decodability as an indicator of task vector effectiveness.** Through mechanistic interventions, we further examine how TD represents the quality of the task vector formation – how effective injecting the task vector is at triggering the desired ICL behavior. By patching layer outputs with either mean activations from correctly inferred latent concepts (*positive intervention*) or the Null class (*negative intervention*), we measure their impact on task performance. As shown in Figure 20, tasks with highly separable representations, such as Noun and Verb in POS tagging and AND and OR in bitwise arithmetic, exhibit significant improvements from positive interventions (up to $\sim 14\%$) and noticeable degradation from negative interventions (up to $\sim 15\%$).

In contrast, tasks with overlapping representations, such as XOR, XNOR, Adjective, and Preposition, show limited sensitivity to interventions, with positive interventions improving performance by only $\sim 2\%$ and negative interventions causing a modest $\sim 6\%$ drop. These differences suggest that when task encoding is done correctly and the representations are well-separated, the task vectors are more effective at guiding decoding processes, as they can easily map distinct concepts to their respective downstream algorithms. Conversely, overlapping representations hinder the ability of the task vectors to reliably trigger distinct decoding algorithms, reducing their overall effectiveness.

Overall, these findings demonstrate that task decodability is a direct reflection of task vector quality. This implies that, when the model becomes more adept at a task, the task becomes correspondingly easier to decode from their representations. We further assess this hypothesis in the next section.

### 4.2. Task Decodability Predicts ICL Task Performance

> **Hypothesis 2**: Quality of task encoding-decoding is predictive of ICL performance.

We now investigate the second hypothesis of whether the quality of task encoding-decoding is predictive of downstream ICL performance. If the model is conditionally applying a decoding algorithm by first inferring the latent task, the quality of the latent task encoding (measured by TD score) and ICL task performance should be closely correlated. To this end, we analyze the relationship between TD and test accuracy in Figure 4. In both datasets, we see that, higher TD scores correspond to better performance on

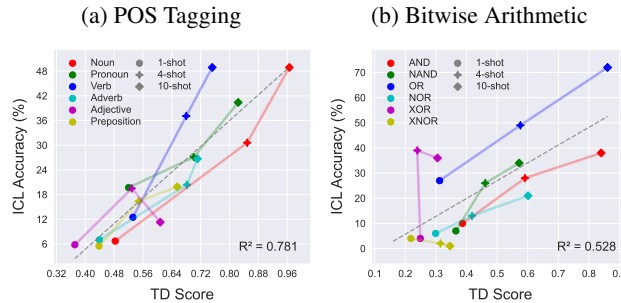

*Figure 4.* **TD score vs ICL performance** in Llama-3.1 8B. We observe a positively correlated trend across most tasks. The grey dashed lines are linear lines of best fit. These results suggest that the accuracy of task encoding is closely coupled with downstream ICL performance.

the respective tasks. Notably, referring back to Figure 3, we again remark that the representations of some classes (Adjective and Preposition in POS tagging and XOR and XNOR in bitwise arithmetic) are mapped to those of the Null class. We notice that this set of classes whose representations overlap with those of Null generally have low task performance and do not improve as much as the others given more demonstrations. We conjecture that the model does not accurately encode the latent tasks of those that are overlapped with the Null class representations.

We also test the generality of our hypothesis that TD predicts ICL performance across different model families and scales. We perform the same analysis on Gemma-2 2B, 9B and 27B (Google, 2024) and Llama-3.1 70B and present the results in Figure 21 in the Appendix G.3. These results demonstrate that the correlation between TD and ICL performance is robust across models and tasks. Interestingly, in all of the Gemma-2 family and Llama-3.1 70B models, Noun, Pronoun, and Verb show the clearest signs of task encoding and decoding behavior, as we saw in the Llama-3.1 8B model. In the bitwise arithmetic task, AND, NAND, OR, and NOR (classes that showed the strongest encoding-decoding behavior in Llama-3.1 8B), also show the strongest signs of task encoding-decoding behavior across all of these models. Given that many LLMs are trained on similar sources of pretraining data (Soldaini et al., 2024; Gao et al., 2020) (CommonCrawl, Wikipedia, etc.), we conjecture that the models may have learned similar task vector mechanisms for these concepts (Huh et al., 2024).

### 4.3. Studying the Emergence of Task Encoding and Decoding in LLM Pretraining

Another natural question is whether the simultaneous emergence of ICL ability and task encoding observed in synthetic experiments in Section 3.3 also occurs during LLM pretrain-

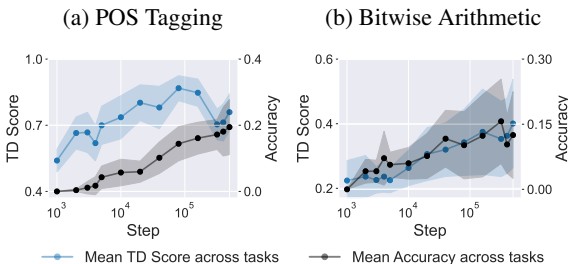

(a) POS Tagging    (b) Bitwise Arithmetic

— Mean TD Score across tasks    — Mean Accuracy across tasks

*Figure 5.* **TD scores and ICL accuracy during OLMo-7B pretraining** (Groeneveld et al., 2024), averaged across POS tagging and bitwise arithmetic tasks, evaluated using 4-shot prompts on 1000 test examples per task. The shaded regions represent the standard deviation across the tasks.

|  | POS Tagging | | Bitwise Arithmetic | |
|---|---|---|---|---|
|  | Avg. TD | Avg. Acc. | Avg. TD | Avg. Acc. |
| Pretrained | $0.68 \pm 0.11$ | $25.5 \pm 6.7$ | $0.43 \pm 0.13$ | $26.2 \pm 15.5$ |
| FT (first 10) | $0.95 \pm 0.02$ | $69.4 \pm 14.2$ | $0.85 \pm 0.11$ | $90.2 \pm 15.0$ |
| FT (last 10) | $0.68 \pm 0.11$ | $33.9 \pm 12.3$ | $0.43 \pm 0.13$ | $66.3 \pm 9.01$ |
| Prompting | $0.97 \pm 0.02$ | $34.4 \pm 10.3$ | $0.90 \pm 0.10$ | $57.2 \pm 30.1$ |

*Table 1.* **Average TD scores and task accuracies with finetuning and prompting** for POS Tagging and Bitwise Arithmetic.

ing. Since large-scale pretraining studies are computationally infeasible, we leverage different training checkpoints of OLMo-7B (Groeneveld et al., 2024) to investigate the relationship between TD and ICL task performance during pretraining. As shown in Figures 5 and 6, increases in TD scores closely align with gains in ICL accuracy. Interestingly, unlike the synthetic setup, this progression is more gradual. This suggests that task encoding and decoding for natural ICL tasks emerge more gradually during pretraining, likely because LLMs are simultaneously learning a diverse range of tasks, making the training dynamics more intricate to disentangle and comprehend. Thus, further investigation into how task-specific representations evolve during pretraining is warranted.

**Generalization to Recurrent Language Models**    Recent work shows that recurrent language models, such as LSTMs (Xie et al., 2021) or Mamba (Gu & Dao, 2023), exhibit ICL behavior (Grazzi et al., 2024). To verify whether our encoding-decoding view explains the emergence and effectiveness of task vectors in such models, we performed the same analysis on Mamba 8B (see Appendix G.4). As shown in Figure 22, a strong positive correlation between TD scores and task performance confirms our hypothesis that TD scores predict ICL performance for state-space models as well. This implies that the proposed task encoding-decoding perspective could potentially be generalized across various architectures beyond transformers.

### 4.4. Improving Task Encoding by Finetuning the Early Layers

**Hypothesis 3**: Building on the encoder-decoder framework, finetuning the early layers enhances task encoding and should yield greater improvements in ICL performance compared to finetuning the later layers.

Given our observation that the earlier layers are responsible for task encoding and the latter layers for the decoding algorithms, we hypothesize that finetuning the earlier layers will enhance task representation and improve ICL task performance more effectively than finetuning the latter layers. This approach challenges the common assumption in model finetuning (Wu et al., 2024; Kumar et al., 2022), where the focus typically lies on adjusting the latter layers or the linear head, under the belief that the final layers primarily govern task-specific adaptation. However, our findings reveal that targeting the earlier layers provides a more effective pathway for improving ICL performance.

Finetuning only the last 10 layers results in minimal improvements to task encoding, as these layers do not change the upstream representations. As illustrated in Table 1, the TD scores of the last 10 layers remain unchanged compared to the pretrained model. In contrast, finetuning the first 10 layers significantly improves TD scores, aligning the representation subspaces with the latent concepts and enhancing task-specific abstractions.

This improvement in representation encoding directly translates to better downstream ICL performance. With 4-shot examples, finetuning the first 10 layers outperforms finetuning the last 10 layers by 37% in the POS task and 24% in bitwise arithmetic. Moreover, finetuning the first 10 layers achieves near-perfect accuracy across bitwise arithmetic tasks, except for XNOR, where overlapping representations with Null limit further improvement.

### 4.5. Investigating the Effect of Prompting on Task Encoding and Decoding

**Hypothesis 4**: Prompting enhances TD by providing a stronger learning signal for task inference, and thus improves ICL performance correspondingly.

In previous sections, we showed that in-context examples and finetuning both improve task encoding and hence ICL task performance. Prompting (Liu et al., 2023) is also a simple and common method to improve a model's performance. In this section, we experiment with prompting to see how

it changes TD along with performance. As shown in Table 1, prompting in fact improves the task encoding and performance simultaneously. Together with the previous results, it suggests that enhancing task encoding may be the unifying principle through which the learning signal materializes in the model's representations across different strategies (e.g., in-context examples, finetuning, and prompting). We detail the setup and results in Appendix I.

## 5. Discussion

Our insights on the origins and mechanism of task vectors have several implications in light of recent works on understanding the mechanics of ICL (Mittal et al., 2024) and activation-steering methods (Bürger et al., 2024; Panickssery et al., 2024; Marshall et al., 2024).

**Why do models succeed at some ICL tasks, but not others?** It is yet puzzling how to categorize the types of ICL tasks LLMs can and cannot solve (Qiu et al., 2023; Dziri et al., 2023). An intuitive explanation is that the model can effectively encode the tasks frequently seen during pretraining (Razeghi et al., 2022; Li et al., 2024). In our experiments, we observe patterns consistent with this conjecture, where AND and OR, the more common logical operators in language, were encoded more accurately. However, under our proposed two-stage mechanism, we show the bottleneck in ICL tasks can exist in both levels of task inference and subsequent decoding algorithms. Therefore, even if the model already learned the algorithm for a task, if the model cannot clearly distinguish the latent concept from the inputs, it will fail, and vice versa.

**Does learning the right latent variables help?** Mittal et al. (2024) investigate whether explicitly modeling the latent variables in ICL outperforms implicit learning through ordinary autoregressive training with transformer. They draw the counterintuitive conclusion that explicit modeling does not enhance performance, albeit not worse; the underlying reasons for which remain unclear. In our work, we explain this specific observation by analyzing the extent to which implicit modeling (standard transformers) captures the true latent variables. Our findings show that transformers can inherently encode these latent variables without explicit regularization. Therefore, we propose that the comparable performance between explicit and implicit models arises not because modeling the latent variables is unhelpful, but because both types of models effectively learn them.

**Limitations.** A limitation of our work is that the experimental setup used in this study does not encompass tasks that require multi-step reasoning (Clusmann et al., 2023; Zupan et al., 1999; Hosseini et al., 2024). Although we analyze the task encoding-decoding mechanism with varying levels of complexity in Appendix D.4, further studies are essential to apply our findings and insights to the real-world. Another limitation stems from our proposed TD metric. Since we measure the separability from one task to the others, for the measure to be meaningful, the distribution of tasks on which TD is computed needs careful design to consist of semantically similar, confusing tasks.

## Impact Statement

This paper presents work whose goal is to advance the field of Machine Learning. There are many potential societal consequences of our work, none which we feel must be specifically highlighted here.

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

# A. Investigating Predictability of ICL Task Performance in Large-Scale Pretraining

Since it is computationally infeasible to conduct large-scale pretraining studies, we leverage the different training checkpoints for OLMo-7B (Groeneveld et al., 2024) to investigate the relationship between concept decodability and ICL task performance on POS tagging. Interestingly, as shown in Figure 5, we observe a correlated emergence of the two variables. This analysis shows that the coupled emergence of concept encoding and decoding algorithms may also hold in large-scale pretraining. However, this warrants further investigation, since we do not fully understand the training dynamics of a LLM.

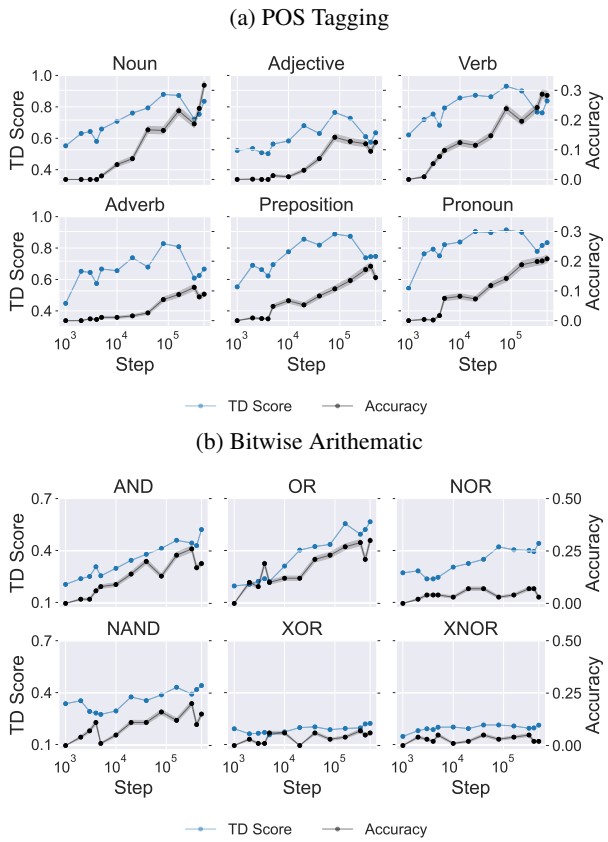

*Figure 6.* Test accuracy and TD scores for POS tagging and bitwise arithmetic tasks across OLMo-7B (Groeneveld et al., 2024) checkpoints (1000-500000 steps), evaluated using 4-shot prompts on 100 test examples per task.

# B. Task Encoding

In this section, we formally define the Task Encoding and Decoding.

**Definition B.1** (Task Encoding). Let $M$ be a transformer model, $\mathcal{Z} = z_1, z_2, \ldots, z_n$ be a set of latent tasks, and $D$ be in-context examples with arbitrary length $K$. A *task encoding* is an internal mapping $E : \mathcal{D} \to \mathbb{R}^{d_{emb}}$, where $\mathbb{R}^{d_{emb}}$ is the intermediate representation over the model's $d_emb$-dimensional embedding space.

**Definition B.2** (Task Decoding). Given a transformer model $M$ with concept encoding $E$, a *task decoding* is a transformer's behavior that there exists a simple function G that can recover the original latent concept and condition the algorithm:

$$G : \mathbb{R}^{d_{emb}} \to \mathcal{Z}$$

ICL performance of given $z$ is related to how well the decoder $G$ can infer the original latent variable $z$. To quantify this, we introduce the notion of *decodability*. For any given decoder, we define decodability as follows:

**Definition B.3** (Decodability). For a given decoder $G : \mathbb{R}^{d_{emb}} \to \mathcal{Z}$ and a specific latent variable $z$, the decodability

measures how accurately the correct latent variable is inferred from representations. Representation is distributed as $E(z, D)$, and the inferred latent variable is $\hat{z} \equiv G(E(z, D))$.

1. **One-hot Accuracy**: $A_{\text{1-hot}}(z) = E[\mathbf{1}[\hat{z} = z]]$

2. **f-divergence**: $A_f(z) = D_f(\hat{z} \parallel z)$, where $f$ is some f-divergence metric

In our study, we employ the one-hot accuracy metric with a kNN classifier to report task decodability.

## C. Activation Patching

We perform the activation patching procedure in the synthetic and natural experiments as follows. We collect representations $\{h_i\}_{i=1}^n$ at selected position and layer for set of in-context examples $\{(x_i, y_i)\}_{i=1}^n$ by passing the prompt through the model and recording the activations at a specified layer after $x_i$. We then average these activations to obtain a single "task" representation $h_{\text{task}} = \frac{1}{n} \sum_i h_i$. For a new query $x_{\text{query}}$, we add $h_{\text{task}}$ into the same layer's activations—effectively replacing the representation after $x_{\text{query}}$—and let the model continue forward.

## D. Synthetic ICL Experiment

### D.1. Theoretical Error Bounds in sparse linear regressions

It is known that transformers can achieve Bayes-optimal solutions for linear regression problems by implementing least-squares solutions on the prior of weight sampling (Garg et al., 2022; Raventós et al., 2024). The least-squares estimation of linear regression with a Gaussian prior for task weights can be performed using ridge regression. In the presence of sparsity, the least-squares solution can be obtained through lasso regression with optimal weight searching. The error bounds of our task depend on whether the underlying basis is discovered by the model. We consider two extreme cases:

1. If the model is incapable of inferring any basis in $\mathcal{B}$, it would perform a $D$-dimensional regression with $r$-sparsity, where $D$ is the total dimension and $r$ is the number of non-zero elements.

2. If the model is capable of inferring the basis in $\mathcal{B}$, it can perform an $r$-dimensional regression adjusted for the corresponding non-zero elements of the inferred basis. In this case, the model could benefit from the tighter $r$-dimensional regression bound.

The possibility of diverse algorithms and corresponding error changes enables us to track the Bayesian inference behavior of the model in a more detailed way. In the following results, we indeed observe a transition from $D$-dimensional regression to $r$-dimensional regression, accompanied by changes in the representations of tasks for each basis.

### D.2. Experimental Details

**Mixture of Sparse Linear Regression.**    We adapt the conventional linear regression setup from (Garg et al., 2022; von Oswald et al., 2023) to create latent bases $\boldsymbol{B}$ that we can interpret far more easily than $W$. We study this setting with $D = 16$ dimensional with up to $K = 20$ in-context examples. Each $B_i$ has a rank of 4 and is orthogonal with each other. We independently sample $W$ and $x_i$ for each new input sequence from $N(0, \boldsymbol{I_D})$ the noise $\epsilon \sim N(0, 0.01)$. We add the sparsity constraints to the linear regression task to introduce the latent concept of sparsity basis $B$ that is easily interpretable and analyzable in their representations. With the sparsity constraints, we construct the graphical model $B \rightarrow W \rightarrow Y \leftarrow X$. This construction allows us to visualize the representations of each of the bases (latent concepts in this graph) by aggregating the representations across a set of $W$ and $(X, Y)$ pairs.

**Model.**    We use a 12-layer GPT-2 (Radford et al., 2019) architecture transformer, as implemented by HuggingFace (Wolf et al., 2020). This model is parameterized with an embedding dimension of 256 and 8 attention heads and hasa total of 9.5M parameters.

**Training.**    We train the model with a batch size of 128 for 80K training steps. We use the Adam optimizer (Kingma & Ba, 2017) with a learning rate of 1e-4 and betas of 0.9 and 0.9999. We use a MSE loss over the sequence and only compute the losses on the prediction $\hat{y}_i$.

**Evaluation.** We construct a test dataset of 1K samples and evaluate the model on MSE loss for the predictions $\hat{y}_i$ along the sequence.

**Compute.** We use an A100 GPU with 80GB of VRAM. To train these models, it takes about $\sim$ 8 hours.

### D.3. Additional results

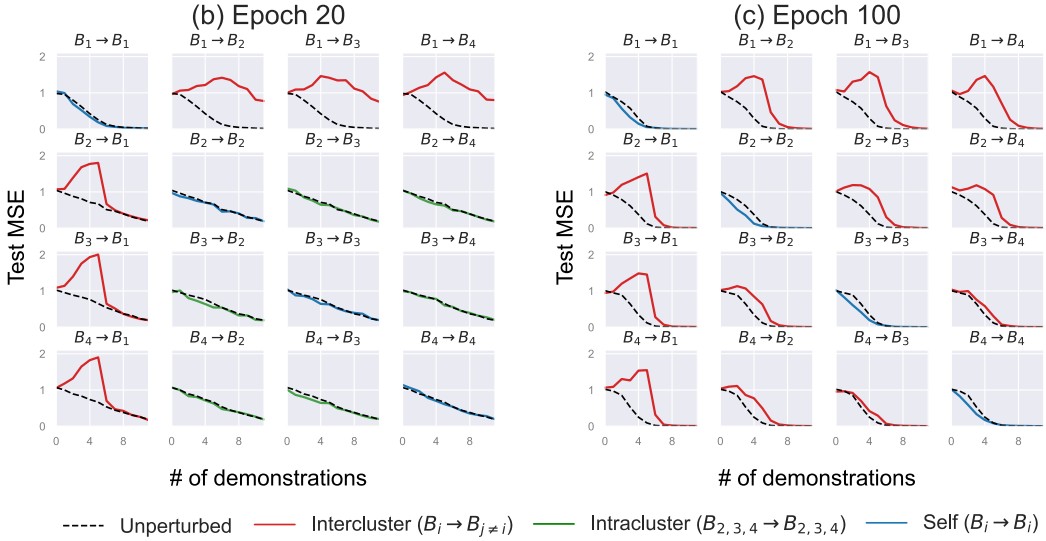

*Figure 7.* Causal analysis by perturbation. On the left are perturbation results at epoch 20, when the latent concepts' representations are semi-separate ($B_1$ and $B_{2,3,4}$). Intracluster refers to $B_{2,3,4}$. At this stage of training when there are only two clusters of representations, there only exists two decoding algorithms as well. On the right are results at convergence, when the latent concepts' representations are fully separable. In this case, each $B_i$ follows a different algorithm and patching the activations of any other basis than itself increases the loss noticeably. On the other hand, self-perturbation improves ICL performance.

**Perturbation analysis to study the causal relation between task encoding and performance over the course of training.**

**Replicate experiments** Here, we run the different seeds of synthetic experiments in Figure 2, and we report the results in figure 8. We observe that a single basis produces distinct loss trajectories for Seeds 1 and 2 as in Figure 2, while Seed 3 demonstrates a consistent loss descent across basis.

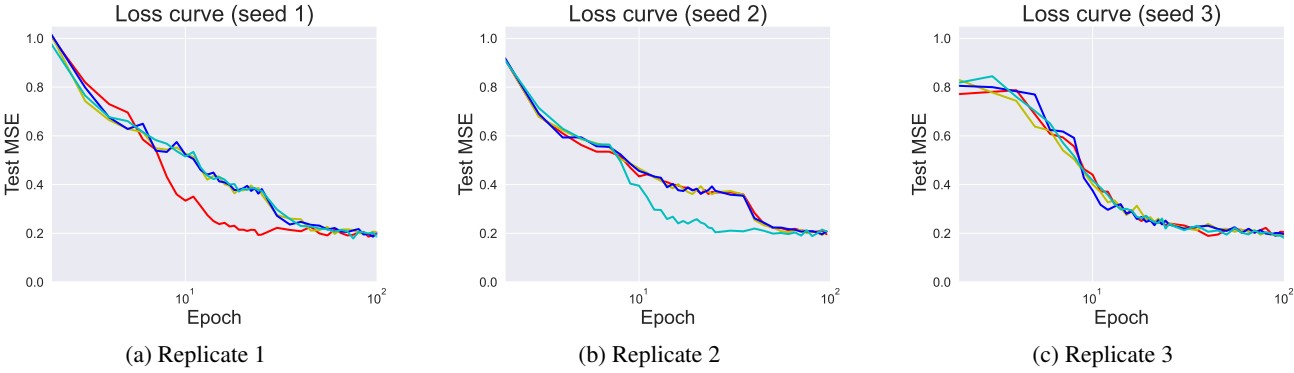

*Figure 8.* Results from three replicates of experiments corresponding to Figure 2. Each subfigure shows the loss trajectory by basis by different random seeds.

### D.4. Additional Analysis on Section 3.3

**TD Over Training.**    We quantified the TD score for the synthetic experiments shown in Figure 2 , with the results presented in Figure 9 and Figure 10. The TD scores for Basis 1 effectively capture the separation of representations observed at (a). An increase in TD scores correlates with a corresponding drop in MSE, as seen in Figure 2, supporting our hypothesis that the TD score can serve as a predictor for the predictability of TD.

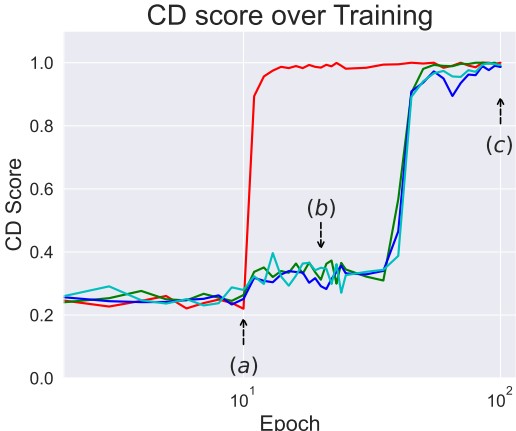

*Figure 9.* TD score of synthetic experiments in Figure 2 over training. (a), (b), (c) denote the same training points in Figure 2.

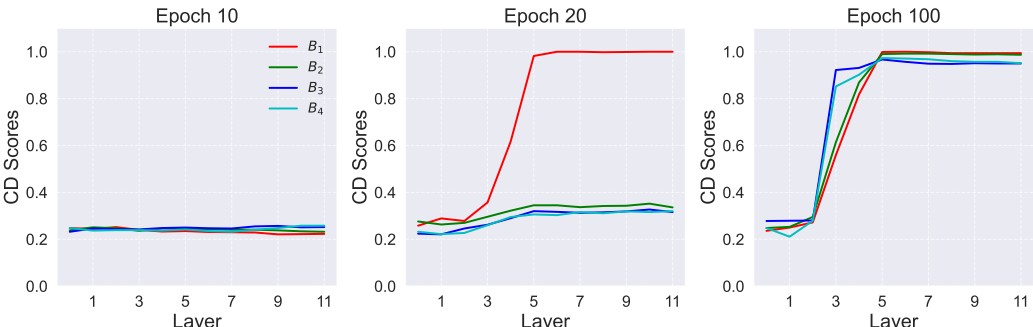

*Figure 10.* TD score across layers at epoch 10, 20, 100 from the synthetic experiment in Figure 2.

**UMAP Over Training.**    To analyze how the representations evolve over training across the different layers in the sparse linear regression task, we visualize the UMAP of the representations in Figure 11. We see that concept encoding, the separation of representations by concept, starts to appear at epoch 20 and is only clearly observed from layer 5. Note that the layer index in the figure starts at 0, so layer 4 in the plot equals to what we call layer 5. At convergence, each of the concepts' representations becomes separated from layer 5 and later.

## E. Increasing complexity in synthetic experiments

### E.1. Experiment - More Orthogonal Bases

We conduct an experiment with 6 orthogonal bases, each spanning 4 dimensions out of 24 total bases. Similar to Figure 1, we observe distinct loss curves over the bases, coupled with clear separation in the representations (see Figure X). Importantly, we observe that basis 6 is learned first (after around 100 epochs), and basis 2 is learned second (after around 200 epochs),

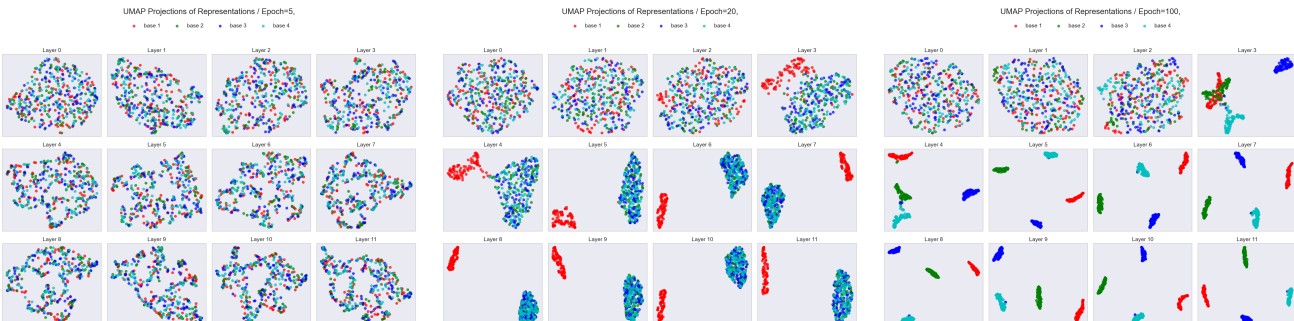

*Figure 11.* UMAP visualization of representations across the layers over training in the synthetic sparse linear regression task. We visualize the UMAP at epochs 5, 20, and 100 across all the layers. Note that the plot uses zero-based indexing, but we use one-based indexing to refer to the layers in all of the text.

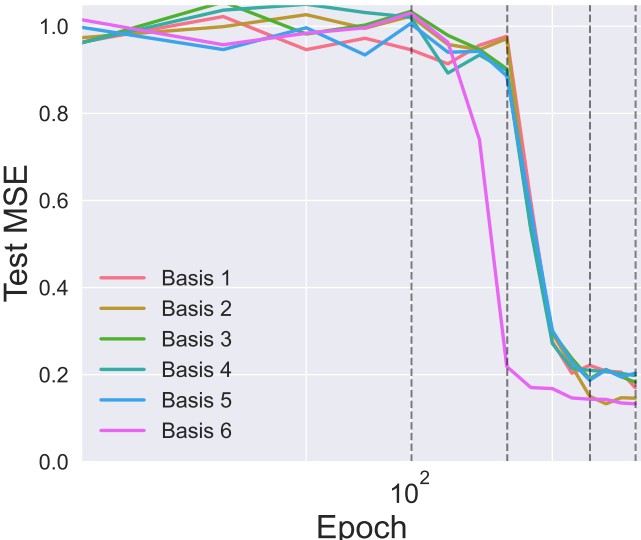

*Figure 12.* Loss curve over training 300 epochs

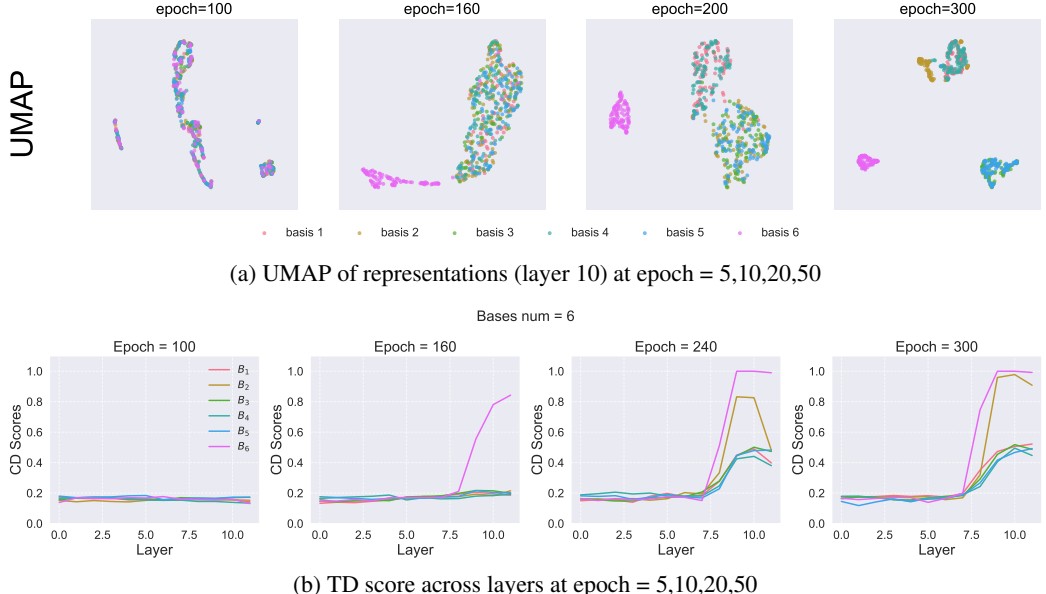

(a) UMAP of representations (layer 10) at epoch = 5,10,20,50

(b) TD score across layers at epoch = 5,10,20,50

*Figure 13.* Experiment - More orthogonal bases analysis

while the other four bases are not distinguished by the model until around 300 epochs. Notably, it requires significantly more epochs for the model to learn each concept compared to the scenario in Figure 1 (which uses 4 bases on 16 input dimensions). Following our intuition, it suggests that learning concepts becomes more challenging as the number of concepts increases. Overall, these results support the idea that our proposed concept encoding-decoding mechanism also holds under more complex settings.

### E.2. Experiment - Overlapping Bases

We conduct an experiment with 8 overlapping bases, where the first 4 bases (Bases 1, 2, 3, and 4) span 8 dimensions, and the remaining 4 bases span the other 8 dimensions (with a total input dimension of 16). Thus, the first four bases have overlap with another and the second bases have overlap with another. In this setup, we investigate the emergence of separation both within overlapping bases (e.g., within Bases 1, 2, 3, and 4) and between the groups (e.g., between Bases 1, 2, 3, 4 and Bases 5, 6, 7, 8), and examine their relation to subsequent ICL performance.

We observe that the loss curve for each base is identical and undergoes a steep descent around epoch 5 (see Figure D-2 in the link). This loss descent coincides with the separation of the two groups of bases by their representations around epoch 5, while bases within the same group remain entangled and unsorted.

These observations suggest several key points. First, the models may not learn to fully separate overlapping concepts, as they can develop shared algorithms to predict the overlapping portions. Second, non-overlapping concepts can be fully separated, which accounts for the significant ICL improvement, as it allows the development of algorithms for orthogonal (non-overlapping) concepts. Third, transformers seemingly learn to classify tasks based on their similarity and associate algorithms at different levels of resolution over the course of training.

## F. Mixture of Different Regression Families Experiment

To further validate the robustness of the concept encoding-decoding framework under more diverse and complex conditions, we conduct experiments with a mixture of regression families—linear, polynomial (degree 3), and sinusoidal regressions. Each regression type represents a distinct latent concept with fundamentally different functional relationships between inputs and outputs.

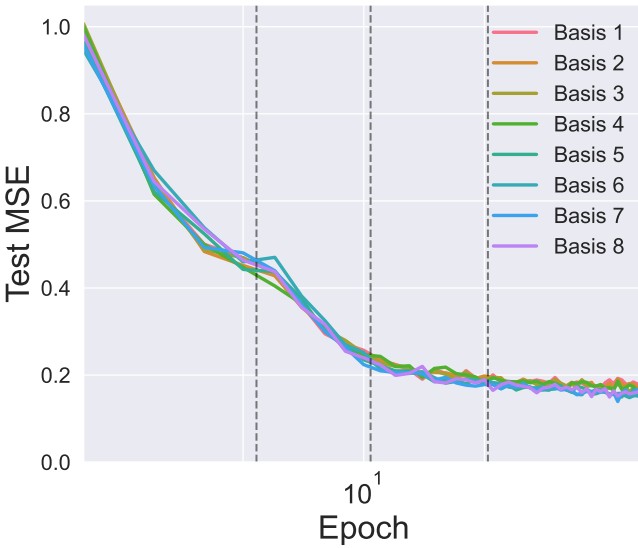

*Figure 14.* Loss curve over training 50 epochs

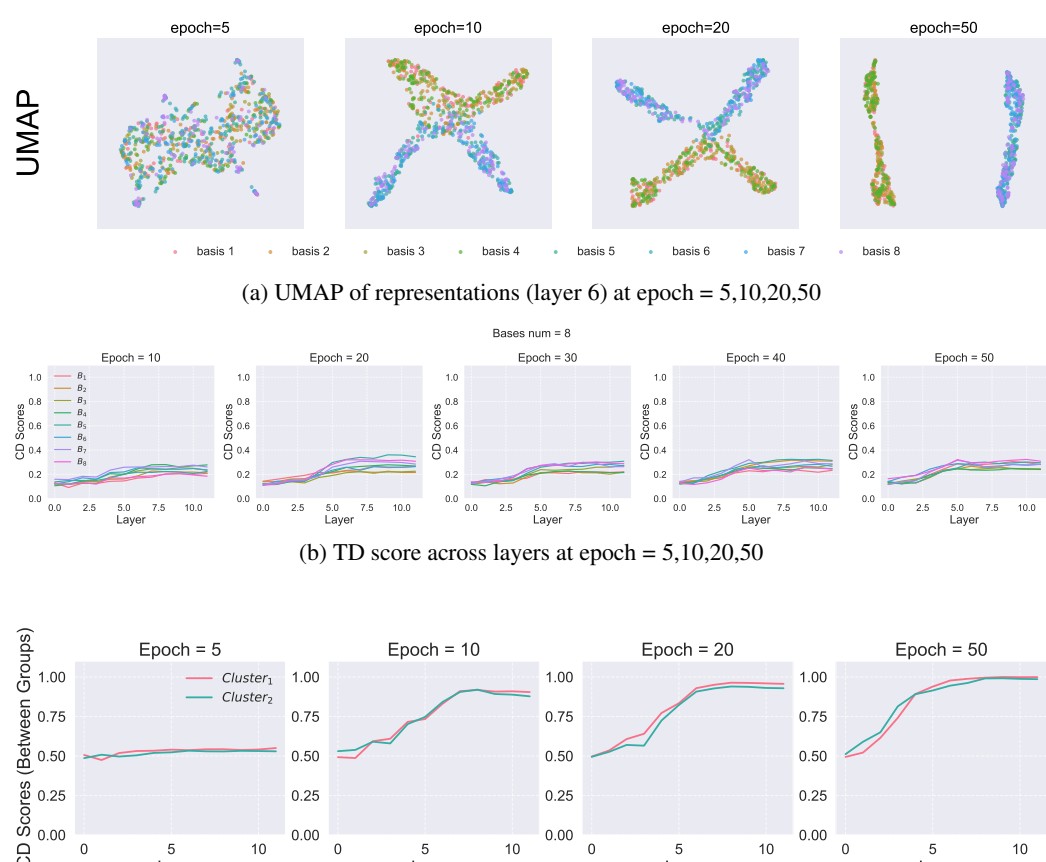

(a) UMAP of representations (layer 6) at epoch = 5,10,20,50

(b) TD score across layers at epoch = 5,10,20,50

(c) TD score between nonoverlapping bases sets(btw basis 1,2,3,4 and 5,6,7,8) at epoch = 5,10,20,50

*Figure 15.* Experiment - Overlapping bases analysis

**Experimental Setup**   We construct datasets with equal proportions of each regression type. Input dimensionality remains consistent with previous setups ($D = 16$), and training parameters follow the synthetic experiment protocol outlined in Appendix D.2.

**Results**   Results shown in Figure 16 indicate clear separability of latent representations corresponding to each regression family after training. Furthermore, we observe significant performance improvements when latent concepts are clearly encoded, confirming that models effectively infer and leverage concept-specific algorithms. This suggests that transformers naturally learn to adaptively switch between qualitatively distinct prediction algorithms based on their internal representations.

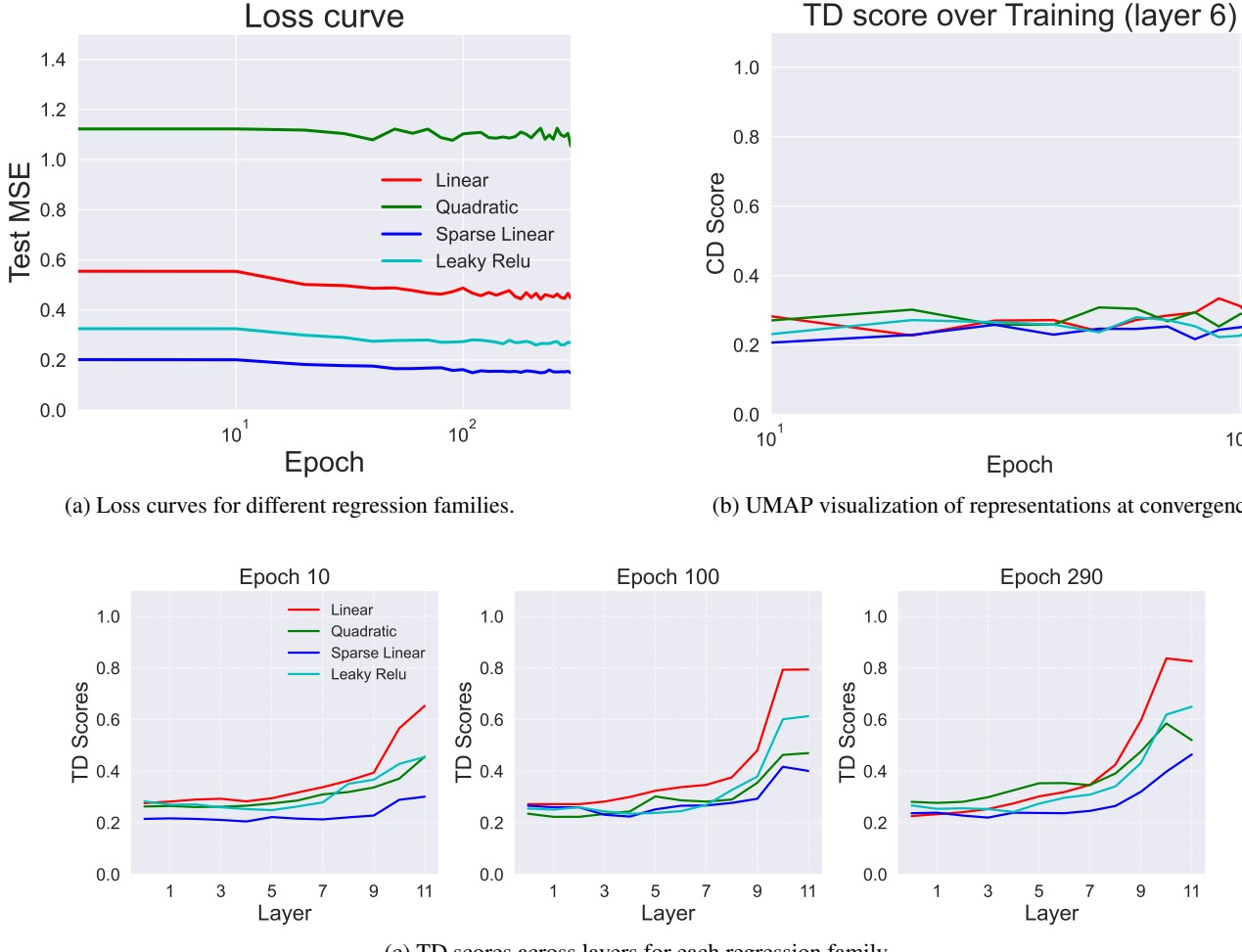

(a) Loss curves for different regression families.

(b) UMAP visualization of representations at convergence.

(c) TD scores across layers for each regression family.

*Figure 16.* **Mixture of Different Regression Families Analysis.** (a) Loss curves highlight performance improvements when latent concepts are clearly encoded. (b) UMAP visualization demonstrates distinct clusters corresponding to each regression family, indicating successful latent concept encoding. (c) TD scores across layers validate the emergence of clear representations for each regression type.

**Verification of Head-Pruning Results**   We validate our claim that distinct latent bases form separate representations by analyzing attention-head pruning effects in synthetic and mixture regression tasks. Attention-head pruning was conducted individually based on magnitude-based importance scores, removing each head one at a time.

Figure 18 shows that attention-head pruning in synthetic sparse linear regression tasks. Changes in mean squared error (MSE) were measured across 100 random sequences per basis (same setup as main manuscript Figure 2). Attention Importance Estimation (AIE) quantifies performance degradation. Layer 4 distinctly maps attention heads to specific bases (e.g., l4h5 for base 0 and l4h3 for base 2), supporting distinct algorithm implementations.

Figure 17 shows that attention-head pruning on mixture regression families. MSE changes assessed across 100 random sequences for each regression type (linear, leaky ReLU, sparse linear, quadratic). Attention heads are notably shared across linear, leaky ReLU, and sparse linear regressions (excluding quadratic), indicating structural similarity. This aligns with the previously observed "common structure" (Kim et al.), confirming structural indistinguishability among these regression variants.

Overall, This confirms that distinct algorithms require dedicated representational capacities, whereas shared algorithms benefit from structural overlap.

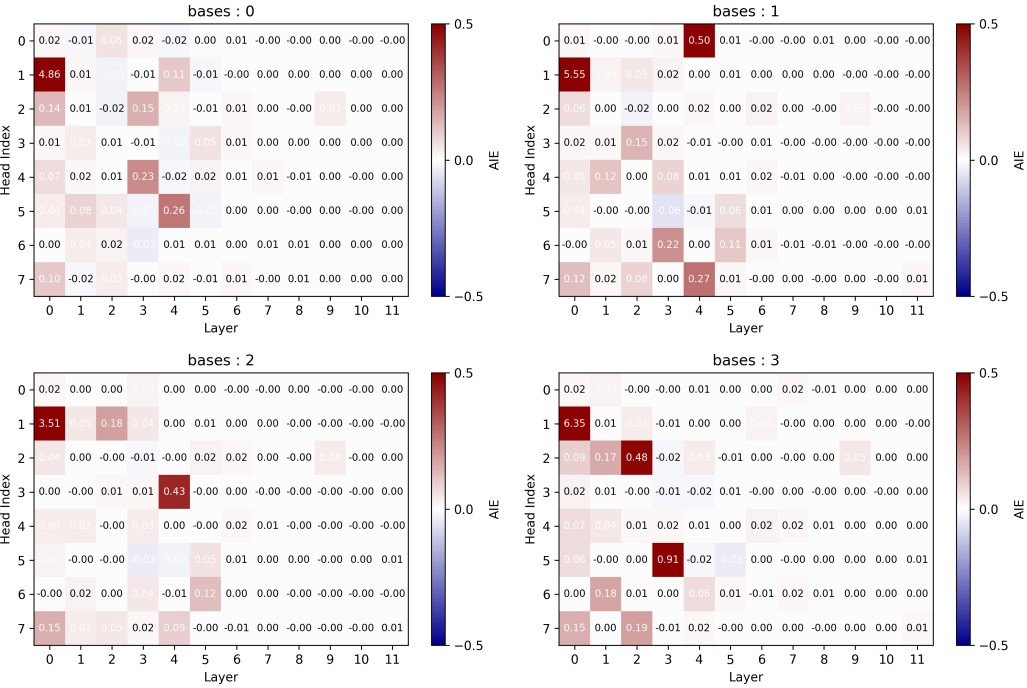

*Figure 17.* **Head-Pruning: Mixture Tasks.** Minimal impact of individual attention-head pruning indicates robustness from shared algorithmic structures.

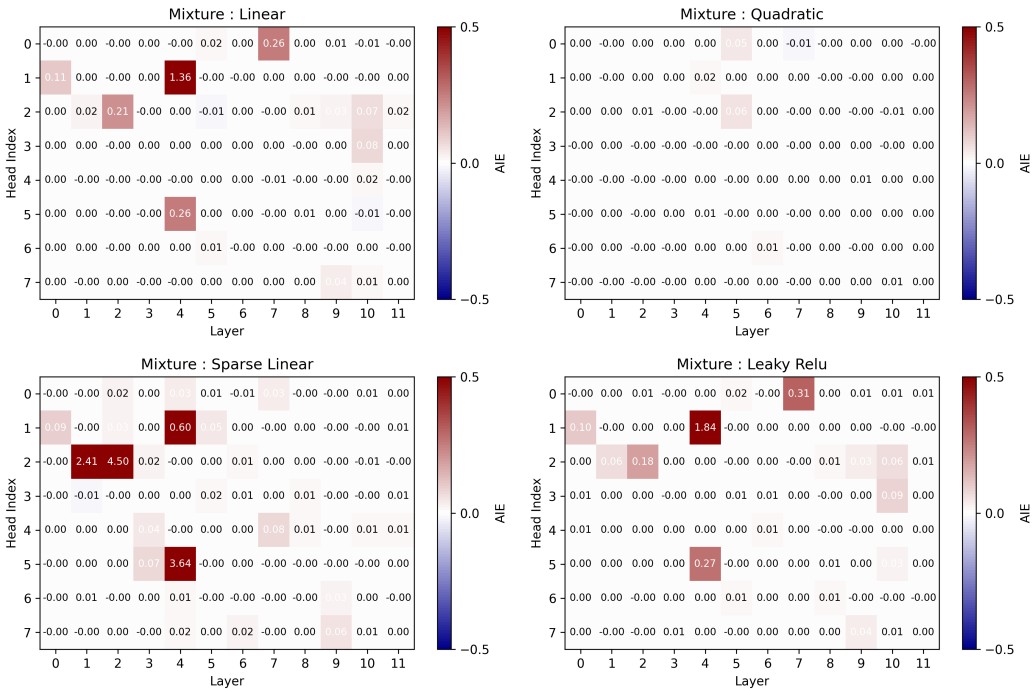

*Figure 18.* **Head-Pruning: Synthetic Tasks.** Individual attention-head pruning significantly affects synthetic task performance, demonstrating distinct basis sensitivity.

## G. Natural ICL Experiments

**Part-of-speech Tagging.** We construct a Part-of-speech (POS) tagging dataset from the English Penn Treebank corpus (Marcus et al., 1994) from the articles of Wall Street Journal. Our POS tags are, Noun, Adjective, Verb, Adverb, Preposition, Pronoun, and Pronoun. We abide by the data-use regulations and, from a total of 4K samples, we filter out sentences that have all 6 POS tags. Then, we split the dataset into a 80-20 train-test split. We evaluate all the models on the test split, and the train split is only reserved for the finetuning experiments.

**Bitwise Arithmetic.** We construct a bitwise arithmetic dataset consisting of 6 different operators: AND, NAND, OR, NOR, XOR, and XNOR. We randomly sample pairs of input binary digits and generate the resulting binary. For training, we construct 10K samples, and, for evaluation, we construct 500 samples.

**Model.** We use a pretrained Llama-3.1-8B model for all of the main natural ICL experiments, if not specified otherwise.

**Training.** For most of the experiments, we do not train the model and only evaluate its ICL performance on the different tasks. However, we only finetune the model in the causal experiments to study the causal relation between the accuracy of concept encoding and ICL task performance. We finetune a model per task family (i.e. POS and bitwise arithmetic). For computationally efficient finetuning given compute constraints, we use LoRA (Hu et al., 2021), a type of parameter efficient finetuning. We set the rank and alpha to be 16 and the dropout to be 0.1. We train the model on a total of 10K samples with the next-token prediction loss. We only backpropagate the losses on the $\hat{y}_i$ predictions.

**Evaluation.** To evaluate the model's ICL performance, we use greedy decoding to generate answers given different number of in-context examples and compute an exact-match accuracy score – whether the generated sequence is exactly equal to the ground truth.

**Compute.** We use an A100 GPU with 80GB of VRAM for training and inference. Training takes $\sim$ 4 hours and evaluation takes $\sim$ 30 minutes for each run.

## G.1. TD score by Layers from Section 4.1

We present the TD scores across the 32 layers of LLaMA 3.1 8B for POS tagging and bitwise arithmetic tasks. We found that TD scores peak at layer 15 for POS tagging and layer 13 for bitwise arithmetic tasks, and we used these layers for measuring TD throughout Section 4. The observation that TD scores peak in the middle layers is consistent with the findings of (Hendel et al., 2023) and (Todd et al., 2023).

(a) TD Scores By Layers

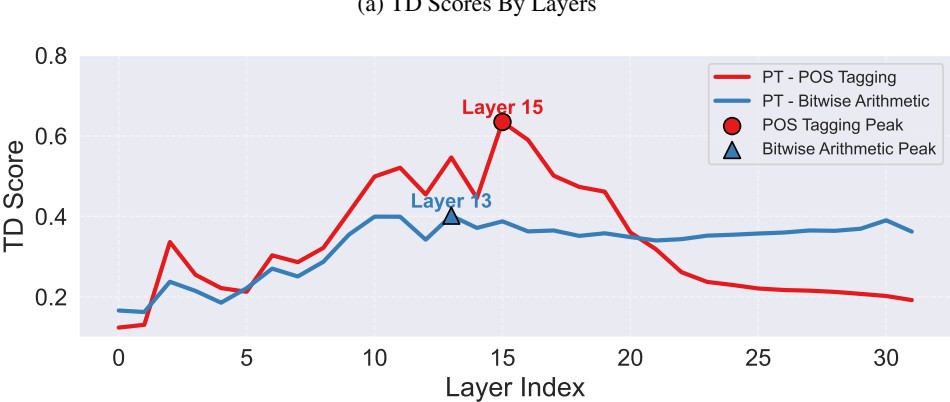

*Figure 19.* TD Scores by layers and number of demonstrations. (a) Mean TD scores across layers for POS tagging and Bitwise arithmetic with 4-shot in-context examples, showing peak decodability in intermediate layers. (b) For POS tagging and (c) for Bitwise arithmetic, TD scores all increase with the number of demonstrations, but the improvement in TD noticeably varies by task.

## G.2. Mechanistic Intervention Study from Section 4.1

We present the results for the mechanistic intervention study probing whether helping or hindering concept encoding improve or degrade activation of corresponding decoding algorithms and whether they are causally related in Figure 20.

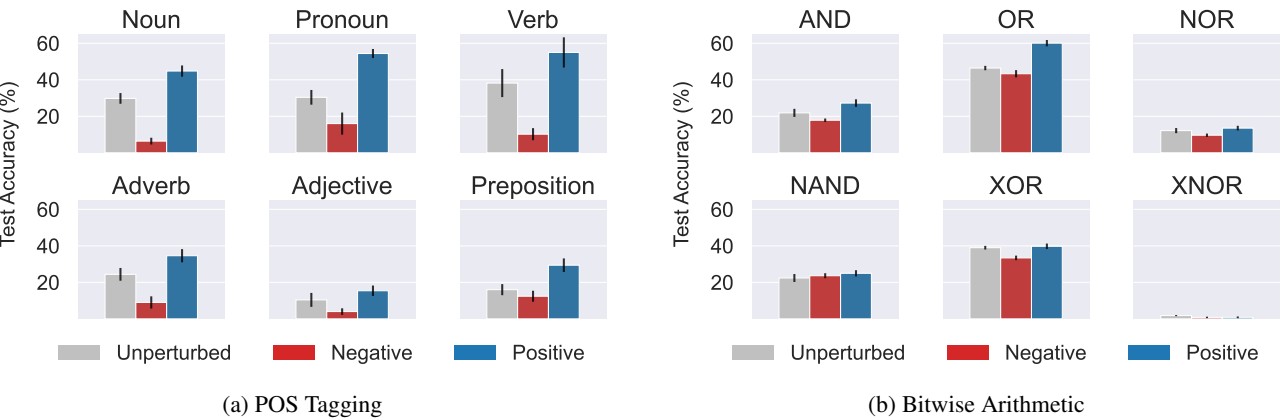

(a) POS Tagging

(b) Bitwise Arithmetic

*Figure 20.* Causal analysis of concept encoding by intervention. We patch the activations of the input with the correct and incorrect latent concept to demonstrate that the inferred concept embedded in the representation can causally improve or degrade performance. We intervene at layers 15 and 13 respectively for the POS and arithmetic tasks. The results show that the performance is causally dependent on the latent concept representations. Error bars represent the standard deviation across five different replicates of experiments.

## G.3. Generalization with Different Model Families and Scales

In both the POS and bitwise arithmetic tasks, we observe a positive correlation between CD and ICL test accuracy across different model families and scales. Interestingly, in all of the Gemma-2 family and Llama-3.1 70B models, Noun, Pronoun, and Verb show the clearest signs of concept encoding-decoding behavior, as we saw in the Llama-3.1 8B model in Figure 4.

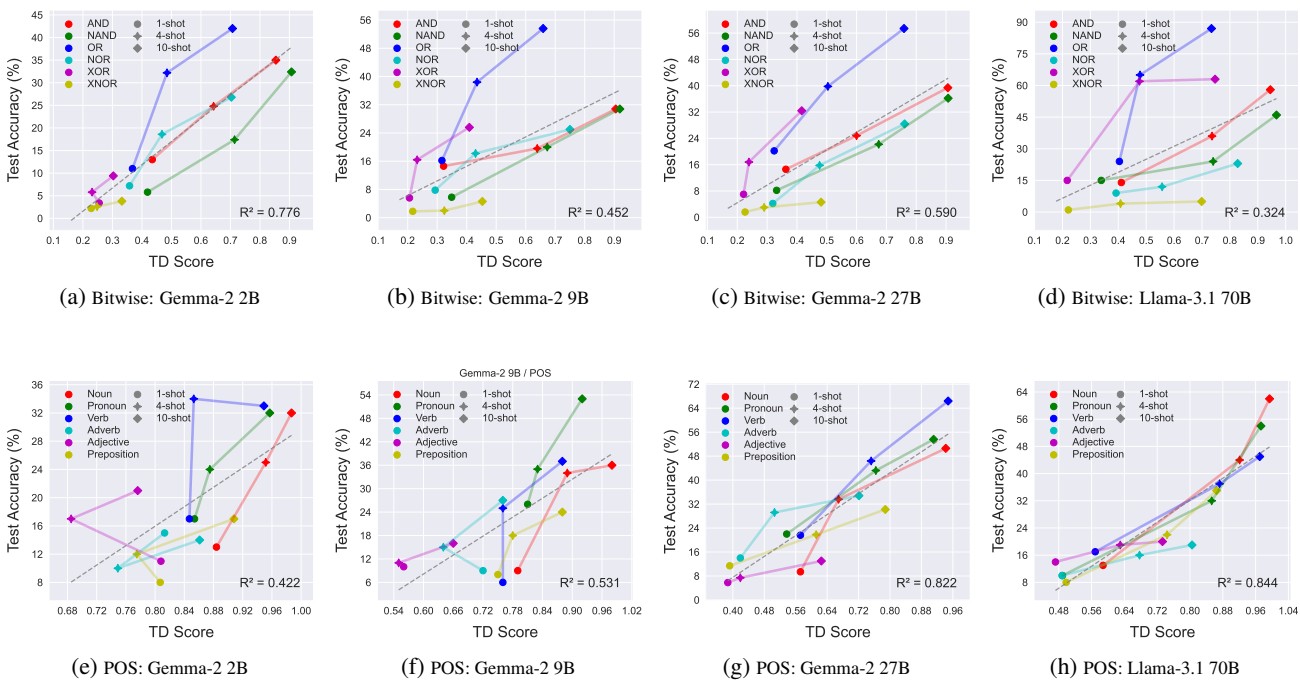

*Figure 21.* TD score vs ICL performance across Gemma-2 models (2B/9B/27B) and Llama-3.1-70B. The positive correlation between TD and ICL performance seen in Llama-3.1-8B generalizes across different models and scales. The grey dashed lines are linear lines of best fit. These results suggest that the accuracy of concept encoding is closely coupled with downstream ICL performance.

In the bitwise arithmetic task, AND, NAND, OR, and NOR (classes that showed the strongest encoding-decoding behavior in Llama-3.1 8B), also show the strongest signs of concept encoding-decoding behavior across all of these models. Given that many LLMs are trained on similar sources of pretraining data (Soldaini et al., 2024; Gao et al., 2020) (CommonCrawl, Wikipedia, etc), we conjecture that the models may have learned similar encoding-decoding mechanisms for these concepts.

### G.4. Extension to Recurrent Neural Network Architectures

To examine whether our task encoding-decoding framework generalizes beyond transformer-based models, we conduct experiments using a recurrent neural network (RNN) architecture. Specifically, we utilize a two-layer LSTM model with 512 hidden units per layer. The training and evaluation setups replicate those used for transformer experiments in Figure 4. Figure 22 shows that the LSTM-based architecture also demonstrates a positive correlation between task decodability (TD score) and in-context learning performance. Although absolute performance metrics differ from transformer-based models, the fundamental relationship between representation clarity and task performance remains consistent. These findings suggest that the task encoding-decoding mechanism is not exclusive to transformers, but also is applicable to sequential neural models more broadly.

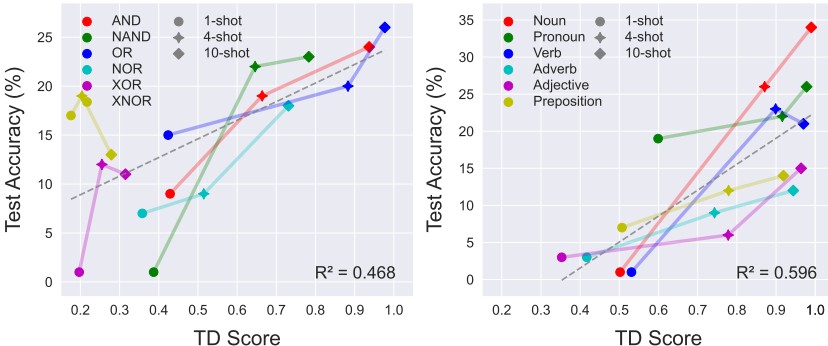

*Figure 22.* TD scores vs ICL perofrmance for Mamba-2 8B model. The positive relationship between TD scores and accuracy in the Mamba-2 8B model suggests that the capacity of TD scores to serve as a proxy for task encoding-decoding processes might generalize to RNN-based language models.

### G.5. Pairwise Concept Decodability Comparison

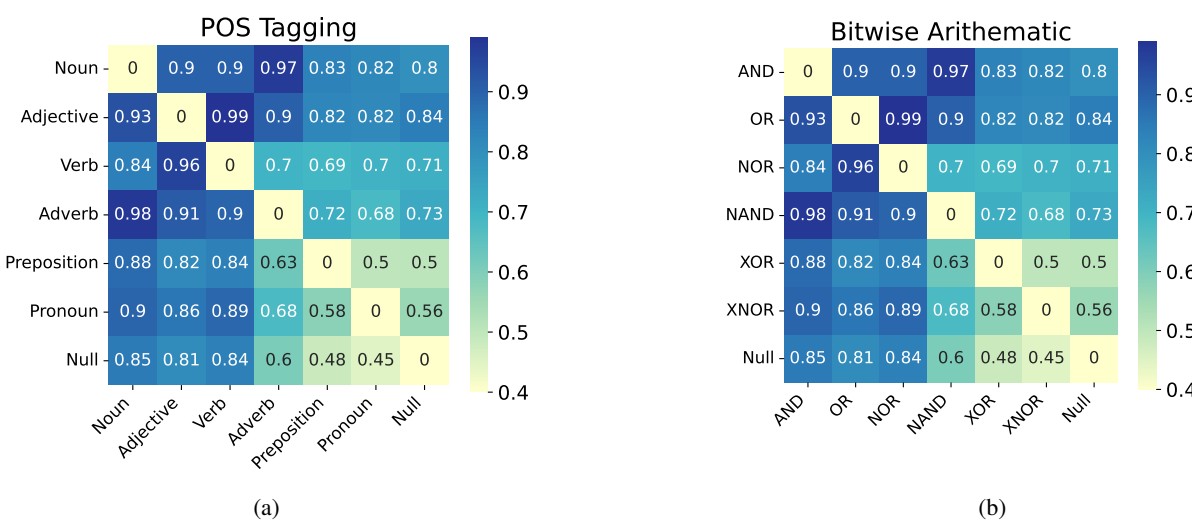

*Figure 23.* Pairwise TD scores for POS Tagging and arithmetic tasks at 4 shot. Pairwise TD scores identifies the clustered tasks

## H. Finetuning Experiments

We visualize the ICL test accuracy changes before and after finetuning the first and last 10 layers of the model on each of the tasks in Figure 24. These results confirm the hypothesis that, contrary to the common practice of finetuning the last layers for classification tasks for instance, finetuning the earlier layers directly improves the task encoding and thus the ICL task performance more than finetuning the latter layers.

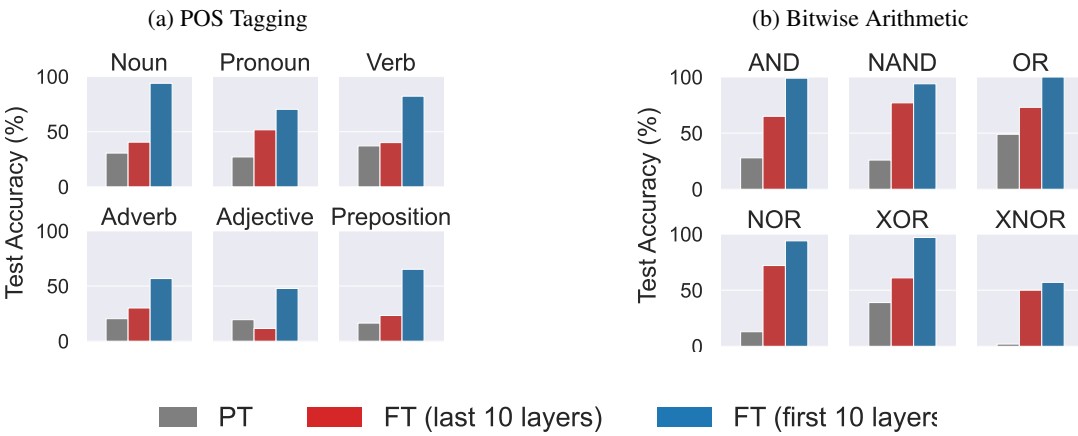

*Figure 24.* ICL test accuracy at 4 shots across 12 tasks in POS and arithmetic after finetuning (FT) the first 10 and last 10 layers. When restricting the model's ability to encode latent concepts in its intermediate representation (finetuning last 10 layers), the model fails to fully align its representations for learning the latent concepts and falls behind the performance of finetuning the first 10 layers.

## I. Prompting Experiments

**Experimental Setup.** To study whether concept encoding is a unifying principle that underlies different mechanisms to improve ICL, we also experiment with prompting. Instead of hiding the concepts and letting the model infer, we include information about the true concept for the examples (e.g., including the true label of AND operator or the instruction of "Find the first noun in the sentence").

**Results.** As discussed in Section 5, we question how prompting may be affecting the concept encoding in increasing task performance. As expected, prompting improves the performance of the model, especially in the bitwise arithmetic experiments. Simultaneously, we observe that the decodability score of the latent concepts also increases drastically. However, we interpret these results with caution because the model may be capturing spurious correlations from the differences in the input distribution. Specifically, the bitwise arithmetic experiments show high decodability even in the beginning layers of the model.

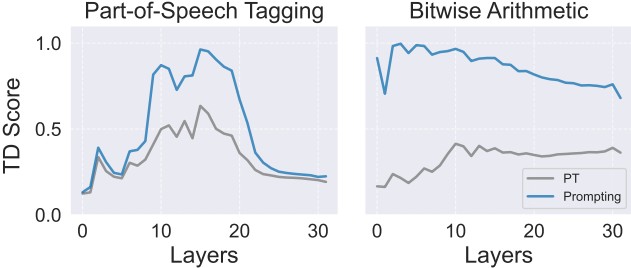

*Figure 25.* TD score across layers for POS tagging and bitwise arithemetic in Llama-3.1-8B for the prompting experiments. We include the true labels of the latent concept (i.e. "Find the first noun in the sentence."). We detail the experimental setup in Appendix I.

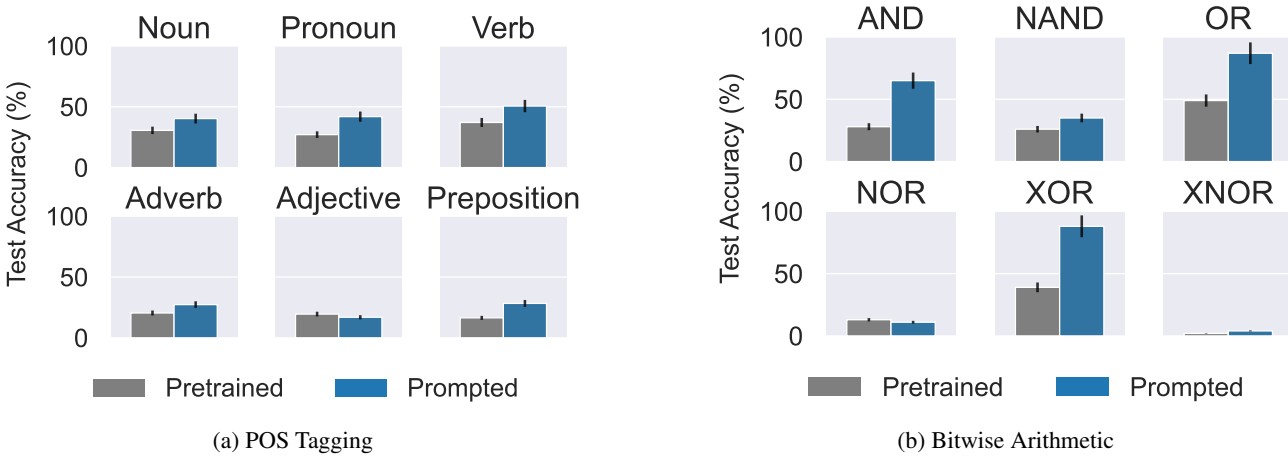

(a) POS Tagging        (b) Bitwise Arithmetic

*Figure 26.* ICL test accuracy across 12 tasks in POS tagging and bitwise arithmetic with prompts containing the true concept (e.g., AND, "Find the first noun in the sentence") of the task.

