# OpenReview forum: "Emergence and Effectiveness of Task Vectors in In-Context Learning: An Encoder Decoder Perspective"
_ICML.cc/2025/Conference — ICML 2025 spotlightposter_

### Official Review · Reviewer_Uvog · 2025-02-14

**Overall Recommendation:** 4

**Summary:**

This work studies the mechanisms underlying the emergence of in-context learning.

The authors show that, on a synthetic dataset composed of a mixture of sparse linear regression tasks, (a) the first layers of a transformer progressively learn to map each task into a separable latent space and (b) the following layers progressively learn task-specific ICL algorithms.

They extend this task encoding-decoding paradigm to real LLMs, showing that some (arithmetic and language) tasks are better encoded than others, which drives their ICL performance. They also propose a task decodability metric, and show that it predicts ICL performance. Finally, they show that finetuning earlier layers improves task encoding hence ICL performance.

**Claims And Evidence:**

The authors make great work at demonstrating the emergence of the task encoding and task decoding mechanisms during training. The synthetic experiments are well-designed and clearly executed
  - Figure 2 clearly shows that task-specific latent separation emerges through training
  - The patching experiment shows that the decoding procedure is task-conditioned.

They also do a good job at showing, on real LLMs, that
  - some tasks are better encoded than others (using UMAP visualization and task decodability)
  - the best-encoded tasks have higher ICL performance
  - since task encoding happens in the first layers, finetuning them boosts ICL performance

**Essential References Not Discussed:**

The authors should cite this paper, which shows the existence of “induction heads” in transformers:
Olsson, Catherine, et al. "In-context learning and induction heads." arXiv preprint arXiv:2209.11895 (2022).

The paper below showed a similar finding to the paper: in a Bayesian sequence model, in-context learning is driven by the existence of task-specific representations
Swaminathan, Sivaramakrishnan, et al. "Schema-learning and rebinding as mechanisms of in-context learning and emergence." Advances in Neural Information Processing Systems 36 (2023): 28785-28804.

**Experimental Designs Or Analyses:**

I really enjoyed the synthetic and the real experiments.

As I was wondering why the main Figure 2 shows the representation of layer 5, I would encourage the author to mention their finding from Figure 11 in the main text: the separation of representations by concept is only clearly observed from layer 5.

**Methods And Evaluation Criteria:**

The methods and evaluation criteria all make sense.

Some comments:
(1) I would encourage the author to detail their patching procedure (line 216).
(2) I understand the idea behind task decodability, but the definition is not precise. See my questions below.

**Other Comments Or Suggestions:**

The authors should add in the limitation section that, in their simple framework (with mixture of sparse linear regression tasks) bases with shared support are not separated in the representation space, which limits their ICL performance.
In contrast, LLMs seem to be able to perform ICL reasonably well on "tasks with shared support" (e.g. acting as famous person P1 vs. P2).

**Other Strengths And Weaknesses:**

The paper is very well written.

**Questions For Authors:**

(1) How many in-context example do you give to the TestMSE loss in Figure 2[left]?

(2) Why does self-pertubation improve performance at convergence (Line 206, right)?
I would imagine that the model knows how to separate the basis, so it would not benefit from patching.

(3) In the definition of the TD score, how do you define the set of representations over which you select the neighbors? Also what is N=100?

**Relation To Broader Scientific Literature:**

The emergence of task encoding (in the earlier layers) and task decoding (in the late layers) in LLMs, and the fact that task encoding drives ICL performance, is an important research problem. The fact that the authors show their findings on synthetic and real tasks makes their results very valuable to the community.

**Theoretical Claims:**

The paper does not contain theoretical claims.

---

> ### Author Rebuttal · Authors · 2025-03-31
>
> Thank you for your detailed and thoughtful review. We greatly appreciate that you find “the authors make great work at demonstrating the emergence of the task encoding and task decoding mechanisms” and that you consider “the synthetic experiments … and the real LLM experiments to be well-designed and clearly executed.” We also appreciate your suggestion to add detailed clarifications and cite relevant works such as Olsson et al., which provided valuable feedback for improving our manuscript.
>
> We now address the specific questions that the reviewer raised their review.
>
>
> > Provide precise definitions of the Task Decodability and patching procedure
> >
>
> **Common Point #3 Raised by Reviewer RjY9, UVog**
>
> We define the patching procedure and task decodability more precisely below and will update the manuscript for the camera ready accordingly.
>
> 1. **Task Decodabilty**
>
> Let $\mathcal{T}$ be the set of latent tasks. For each task $z \in \mathcal{T}$, sample N=100 datapoints $\{(x_i, y_i)\} _ {i=1}^N$ and collect the intermediate representations $\{(h_i)\} _ {i=1}^N$ from a chosen layer (e.g., the token embedding immediately after $x_i$). Label each representation $h_i$ with the corresponding task $z$. This yields a set
> $\mathcal{S} = \bigcup_{z \in \mathcal{T}} \{ (h_i, z) \mid i = 1, \dots, N \}$
> , over all tasks $z$.
>
> Given a query point $(h_i, z)$, we exclude $(h_i, z)$ from $\mathcal{S}$ (i.e., $\mathcal{S} \setminus(h_i,z)$) and find its $k$ nearest neighbors (with $k=10$) in the remaining set. We then use majority voting on those neighbors’ task labels to produce a predicted label $\hat{z}$. If $\hat{z} = z$, classification for the task label is correct. The TD score at this layer is the fraction of query points classified correctly:
>
> $\text{TD} \coloneqq \frac{1}{|\mathcal{D}|}\sum_{(h_i, z)\in \mathcal{D}} \mathbf{1}\bigl[\hat{z} = z\bigr].$
>
> 2. **Patching Procedure**
>
> We collect representations $\{h_i\} _ {i=1}^n$  at selected position and layer for set of in-context examples $\{(x_i,y_i)\} _ {i=1}^n$  by passing the prompt through the model and recording the activations at a specified layer after $x_i$. We then average these activations to obtain a single “task” representation $h_{\text{task}} = \frac{1}{n}\sum_i h_i$. For a new query $x_{\text{query}}$, we add $h_{\text{task}}$ into the same layer’s activations—effectively replacing the representation after $x_{\text{query}}$—and let the model continue forward.
>
> > Why does the model show maximal representation separation at layer 5 in the figure 2?
> >
>
> Thank you for noting that detail, as it has been a common point of interest for the reviewers (Refer to **Common Point 1**). Indeed, Figure 11 in the appendix shows that the separation of representations by concept emerges most distinctly at layer 5, which is why we focus on that layer in the main Figure 2.
>
> > Suggestions for citing several papers(Olsson, Catherine, et al. (2022) and Sivaramakrishnan, et al.(2023))
> >
>
> Thank you for these suggestions, as they seem very relevant. We will add these to our discussion of related works in the final camera ready.
>
> > Suggestions for adding a limitation about the discrepancy between the simple setup and LLMs for tasks with shared support.
> >
>
> We agree this is a notable limitation of our simpler mixture-of-sparse-linear-regression framework. Perhaps, the large-scale pretraining in LLMs allows them to handle overlapping task supports more effectively, whereas our synthetic setup sometimes struggles to separate representations for tasks with shared support. We will add this in our limitations section for our camera ready version.
>
> > How many in-context example do you give to the TestMSE loss in Figure 2?
> >
>
> The TestMSE in Figure 2[left] is mean squared error over sequence of 20 in-context examples. Specifically, TestMSE $\epsilon$ is given as $\epsilon = E[\frac{1}{20} \sum_{i=1}^{20} \left( y_i - \hat{y}_i \right)^2]$. We will clarify this point in the followup.
>
> > Why does self-pertubation improve performance at convergence (Line 206, right)?
> >
>
> Figure 7 that the reviewer is referring to shows test MSE (y-axis) vs. number of demonstrations (x-axis). Because the task dimension is 4, having fewer than 4 demonstrations leaves uncertainty about the underlying basis. In this case, self-perturbation (patching the task vector) provides extra information about the basis and thus improves performance in that regime. That’s precisely why once the number of demonstrations matches the task dimension (around 4), the model identifies the task basis, and patching offers little to no additional benefit.
>
> > In the definition of the TD score, how do you define the set of representations over which you select the neighbors? Also what is N=100?
> >
>
> We define the set of representations by first sampling 100 datapoints (N=100) from *each task* and collecting their representations. We then compute the TD score by performing kNN classification on these representations.

---

> > ### Comment · Reviewer_Uvog · 2025-04-03
> >
> > I thank the reviewer for carefully answering my questions.
> > I encourage them to add the definitions of the task decodability and of the patching procedure in the camera-ready version.

---

### Official Review · Reviewer_bnoD · 2025-03-08

**Overall Recommendation:** 4

**Summary:**

This paper studies the task vector behavior in the trained-from-scratch and pretrained transformer, demonstrating the following conclusions:
1. When the transformer develops the task encoding (that generates the task vector), it simultaneously generates the corresponding decoding functions.
2. In pretrained LLMs, the well-separation of task vectors correlates with the ICL performance.
3. Finetuning earlier layer helps for ICL performance.

**Claims And Evidence:**

Please see "Experimental Designs Or Analyses"

**Essential References Not Discussed:**

NA

**Experimental Designs Or Analyses:**

In the initial synthetic experiment, the author provides the evidence that one of the basis classes is decoded first compared to the other basis functions. I wonder if the 4 basis functions are equally difficult, why the model chooses to decode the basis function B1 first?

In observation 2, why choose the 5th layer, instead of any other arbitrary layer that is early or intermediate for the transformer? How do you validate that the representation at the 5th layer is strong enough for the later "casual relation between task encoding and performance" experiment?

I think the different encoding progress for different tasks is really interesting; however, it is not clearly shown in any other synthetic tasks except the sparse linear regression. In the pretrained LLM experiment on OLMo-7B, only the correlation between "clear separation between task vectors" measured by TD and the ICL performance is compared, leaving me wondering whether the separations occurred also in the pretraining. As a first step, could you also conduct experiments like Figure 7 for Figure 6 to understand the learned decoding for different tasks?

In section 4.2, the author hypothesizes that the quality of task vector is predictive of ICL performance. However, in the recent preprint [Yang et al. 2025], it seems that the Transformer can solve the synthetic in-context learning task regardless of whether a clear task vector (and thus the accompanying encoding-decoding algorithm) is formed. Would this hypothesis only apply to the pre-trained LLM that is not yet finetuned on the downstream tasks? For instance, in table 1, even the prompting could result in good TD score, however, with worse ICL performance as in the Bitwise Arithmetic task. How to properly support this hypothesis then?

I didn't quite understand the logic behind "finetuning the earlier layer is more effective given the earlier layer is for task encoding". Since the decoding algorithm needs to be developed as well given the task vector, and when the task encoding is not developed, the task decoding is not developed as well, why only learning the task vector by finetuning the earlier layer helps? Why learning task encoding is more important than learning the task decoding? Could you elaborate this further?

Following the question above, I noticed for the bitwise arithmetic task, though layer 13 has the peak performance, it does not differ a lot with the remaining layers (after layer 13) as shown in Figure 16. How do you decide layer 13 is the layer that task vector resides?


[Yang et al. 2025] Yang, Liu, et al. "Task Vectors in In-Context Learning: Emergence, Formation, and Benefit." arXiv preprint arXiv:2501.09240 (2025).

**Methods And Evaluation Criteria:**

Yes

**Other Comments Or Suggestions:**

No.

**Other Strengths And Weaknesses:**

No

**Questions For Authors:**

I appreciate the author trying to develop and explain the behavior of task vector. However, the message of this paper seems to be a bit mixed. The first synthetic experiment demonstrates that the encoding - decoding algorithm emerges at the same time the task vector emerges for certain tasks. However, the later part of the paper focuses on the task vector quality correlating with the ICL performance. I think both are interesting findings, but the flow is a bit confusing to me.

**Relation To Broader Scientific Literature:**

The key contribution of the paper extends the previous literature on task vectors by complementing the existence of task vectors with the hypothesis that the Transformer learns the task vector along with the encoding-decoding algorithm, and that the formation of task vectors helps in-context learning performance.

**Theoretical Claims:**

NA

---

> ### Author Rebuttal · Authors · 2025-03-31
>
> We are grateful for your careful review. You highlight a key point that “the different encoding progress for different tasks is really interesting,” which resonates strongly with our primary objective of illustrating how task encodings develop and shape downstream ICL performance.
>
> Below, we address your comments and questions.
>
> > Why does the model learn to encode B1 first in the synthetic experiment?
> >
>
> Although the four bases are theoretically equally difficult (same rank), random factors like data order and seed can cause the model to decode one basis first. As shown in Fig. 8, different seeds lead to different bases being prioritized.
>
>
>
> > How does our observation explain or contradict Yang et al. 2025?
> >
>
> We agree that the formation of distinct task vectors depends heavily on task formalism (how distinct task are defined). When the tasks share a common structure (e.g., the variant of regression tasks in Yang et al. [1] and Von Oswald et al. [2]), the model has little incentive to learn distinct representations, and simple attention-based algorithm[2] may suffice. This aligns with the Kim et al. [3], who report the underlying common structure between the aforementioned regression tasks.
>
> Our synthetic sparse regression experiments are designed with algorithms that differ across tasks(basis), effectively enforcing a algorithmic diversifications and formation of distinct task vectors(albeit with varying quality). From these experiments, we report that the encoding and decoding algorithms emerge *simultaneously*, and that the quality of the learned encoding (measured by TD score) correlates with ICL performance.
>
> We acknowledge that real-world tasks lie on a spectrum: some share overlapping structures, while others require truly orthogonal algorithms. This naturally places them between the two extremes of having fully separate versus completely shared task representations. Nonetheless, our key takeaway is that **when tasks are sufficiently distinct, the quality of the learned task encodings can serve as useful indicator of ICL performance**.
>
> [1] Yang, Liu, et al. (2025).
>
> [2] Von Oswald, Johannes, et al. (2023).
>
> [3] Kim, Jaeyeon, et al. (2024).
>
> > What is the criterion for layer selection for Fig. 2?
> >
>
> Thank you for highlighting this point—it aligns with a common reviewer concern (Refer to **Common Point #1** in RjY9’s rebuttal). As shown in Fig. 11, concept separation is most distinct at layer 5, which is why we select it in Fig. 2.
>
> > In Table 1, prompting yields a high TD score but poor ICL performance on the Bitwise Arithmetic task. How can this observation be better explained?
> >
>
> While prompting (i.e., modifying only the input text) leads to strong TD scores, it does not always result in correspondingly high accuracy—particularly compared to finetuning cases. As discussed in Appendix G, we interpret this results with caution: high decodability observed in the early layers during prompting may not reflect meaningful task encoding-decoding. Instead, it could stem from **spurious correlations introduced by differences in the input token distribution**. The bitwise arithmetic experiments seem sensitive to these correlations, which can inflate decodability as shown in Figure 21 on the right. Nevertheless, prompting still consistently produces higher TD scores and accuracy than the pretrained model.
>
> > Why does learning the task encoding by finetuning only the early layers improve performance? Why is learning encoding more important than decoding?
> >
>
> That’s an insightful question. Our hypothesis is that, since the LLM is pretrained on vast amounts of data, it often already has the “decoding” circuits (e.g., it can do XOR/AND) and mainly needs the right trigger(encoding signal) to activate them. Thus, directly changing the earlier layers (task encoding) can often be critical. However, for truly novel tasks that the model hasn’t seen (e.g., coding in an esoteric programming language [1]), it may need to learn both the task representation and the decoding algorithm from scratch.
>
> [1] https://en.wikipedia.org/wiki/Esoteric_programming_language
>
> > Two messages on emergence of task vectors and relationship between TD and performance.
> >
>
> Our paper indeed focuses on two key findings: (1) how task vectors emerge during pretraining and (2) the quantification of task vector quality. While these findings may initially appear distinct, they form a cohesive narrative as follows.
>
> **We first study the direct phenomenology of task vector emergence** with the synthetic experiments and establish the foundation of our task encoding-decoding framework. Then, for the second point, we demonstrate **how this framework is practically useful** by using task decodability to predict downstream ICL performance in natural setting—bridging mechanistic insights with applied relevance.
>
> Therefore, we believe the two claims offer a comprehensive encoding-decoding framework for understanding and evaluating ICL capabilities.

---

> > ### Comment · Reviewer_bnoD · 2025-04-05
> >
> > Thanks for taking the time to address my comments. I appreciate the insightful investigation in this paper. I have raised my score.

---

### Official Review · Reviewer_EJ7t · 2025-03-12

**Overall Recommendation:** 3

**Summary:**

This paper investigates the learning mechanisms of autoregressive Transformers in In-Context Learning (ICL), with a particular focus on the emergence of task vectors and their impact on ICL task performance. Building on the Bayesian view of ICL, the authors propose an encoder-decoder analytical framework and empirically demonstrate its validity by examining the evolution of representations during training. The study further reveals that Transformer models develop task representations throughout pretraining, and the quality of task encoding strongly correlates with and effectively predicts ICL task performance.

**Claims And Evidence:**

The theoretical logic is comprehensive, extending from experimental observations in regression tasks to natural language experiments. However, the data sources for some figures are not clearly stated, leaving us puzzled about which data were used for UMAP in Figure 2.
The article, based on a Bayesian view, uses UMAP in regression experiments to demonstrate the geometric structure of the representation space and measures loss using MSE, explaining the emergence of separable representations and coupled algorithmic phase transitions. However, it does not clarify why the fifth layer activation was chosen for substitution or the effects of substituting activations from other layers. Figure 7 effectively illustrates the encoder-decoder framework.

The article proposes the TD score as a basis for selecting important layers, inspired by the background of task vector proposals. However, it is unclear how this differs from previous methods, such as those in (Todd et al., 2023), and whether the principles are consistent, which is a point of confusion for us.

**Essential References Not Discussed:**

No.

**Experimental Designs Or Analyses:**

NULL (I did not reproduce the experiments, but the experimental logic is clear and complete.)

**Methods And Evaluation Criteria:**

The methods are reasonable, drawing conclusions from phenomena in regression tasks to inform understanding of natural language tasks. The loss settings during training and the calculation of the TD score are scientifically sound. The article has guiding significance for subsequent applications, such as using the encoder-decoder framework to characterize task vectors, which can guide the completion of ICL tasks.

**Other Comments Or Suggestions:**

It is recommended to supplement the data sources used in the UMAP method and attempt to discuss whether there is consistency with the specific task vector extraction methods in related articles.

**Other Strengths And Weaknesses:**

**Strengths**

1. The theoretical logic is comprehensive, extending from experimental observations in regression tasks to natural language experiments.
2. This paper systematically elucidates the formation mechanism of task vectors through an encoder-decoder analysis framework and verifies its predictive ability for ICL task performance. This research provides new theoretical foundations and empirical support for understanding the performance mechanisms and optimization methods of large language models in ICL.

**Weaknesses**

1. The specific data sources for some conclusion charts are not clearly stated.
2. The task complexity adopted in the paper is relatively low.

**Questions For Authors:**

- Regarding Figure 2, what specific points were used when applying UMAP? Are the points used for the projections of the four different regression tasks consistent? Specifically, what data were utilized? Was the projection matrix consistent during the projection process?
- Additionally, in line 197 of the article, it is mentioned that the fifth layer activation was chosen. How was the fifth layer selected?
- In Figure 7 (a), why do the solid and dashed lines coincide when the number of samples in the first column increases, but not in the first row?
- In Figure 7, why is the effect eliminated as the number of samples increases?

**Relation To Broader Scientific Literature:**

The calculation of important layers using the TD score aligns conceptually with the identification of critical activations in (Todd et al., 2023). However, the theoretical computation process differs, and the specific connections between the two approaches warrant further exploration.

**Theoretical Claims:**

The paper proposes that the formation of task vectors can be explained through an encoder-decoder framework, and that Task Decodability is a predictor of ICL performance. The theoretical logic is complete, and the description is clear.

---

> ### Author Rebuttal · Authors · 2025-03-31
>
> Thank you for your positive and thoughtful review. We appreciate that you see our “theoretical logic as comprehensive, extending from experimental observations in regression tasks to natural language experiments,” and that you find our encoder-decoder framework provides “new theoretical foundations and empirical support for understanding the performance mechanisms and optimization methods of large language models in ICL.”
>
> Below, we address your comments and questions.
>
> > The task complexity adopted in the paper is relatively low.
> >
> **Common Point #2 Raised by Reviewer RjY9, EJ7t**
>
> We agree that our selected tasks are relatively simple compared to MMLU for example, but we intentionally focus on standard tasks (e.g., Oswald et al., Garg et al., Chen et al.) to rigorously study the emergence and effectiveness of task vectors using clearly distinguishable concepts (e.g., AND/OR in Bitwise Arithmetic, Pronoun/Verb in POS tagging).
>
> For more complex settings in our synthetic setup, we experimented with higher numbers of bases and overlapping/shared bases between tasks (see Appendix D.1 and D.2). Interestingly, we observed that models tend to develop shared algorithms for overlapping concepts while distinctly separating non-overlapping ones—driving significant improvements in ICL performance.
>
> [1] Von Oswald, Johannes, et al. "Transformers learn in-context by gradient descent." ICML. PMLR, 2023.
>
> [2] Garg, Shivam, et al. "What can transformers learn in-context? a case study of simple function classes." NeurIPS 35 (2022): 30583-30598.
>
> [3] Chen, Xingwu, Lei Zhao, and Difan Zou. "How transformers utilize multi-head attention in in-context learning? a case study on sparse linear regression." arXiv preprint arXiv:2408.04532 (2024).
>
> > What is the criterion for layer selection for task vector analysis and how is it related to Todd et al., 2023?
> >
>
> We refer the reviewer to our **rebuttal to Reviewer RjY9 (under Common Points #1)** for how we select the layer based on the highest TD score.
>
> This approach is aligned with Todd et al. 2024. Todd et al. observe that task vectors at different layers result in varying ICL performances (Figure 2c in their paper).  Building on this, we propose TD score to quantify the “quality” of task vectors. We demonstrate in Section 4 that **(1) TD score is *predictive* of downstream ICL performance directly and (2) both finetuning and prompting for ICL can be interpreted under this framework of an encoder-decoder**.
>
> > What data and procedure is used for UMAP in Figure 2?
> >
>
> To generate the UMAP in Figure 2, we randomly draw 100 samples for each basis. Next, we collected $y$-token representations of 5th layer at 20th demonstrations and plotted the UMAP with the parameters of `n_neighbors=15` and `min_dist=0.1`. This specific layer was selected based on the maximum TD score (See Common Point #1 under Reviewer RjY9 and Figures 10 and 11 in the manuscript).
>
> > Minor improvements in supplementary presentation
> >
>
> We appreciate the feedback and will refine the clarity of our supplementary results for the camera-ready version.
>
> > In Figure 7 (a), why do the solid and dashed lines coincide when the number of samples in the first column increases, but not in the first row?
> >
>
> We hypothesize that the phenomenon occurs because the decoding algorithm for B2,B3,B4 may have developed after the 5th layer at epoch 20.  This allows perturbations to B1 introduced at the 5th layer to be corrected as there are additional encoding layers before decoding occurs, when sufficient demonstrations are provided.
>
> In contrast, the decoding algorithm for B1 appears to be implemented directly at the 5th layer, as evidenced by the separated representations visible in the UMAP visualization in Figure 10. Consequently, the model cannot be recover from perturbations from B1 to B2, B3, or B4 at the 5th layer.
>
> > In Figure 7, why is the effect eliminated as the number of samples increases?
> >
>
> This is an example of when **LLMs can recover the latent task against the adversarial task vector perturbation (patching wrong task vector)**. Previous works by Usman et al. [1]. and Chen et al. [2] show that higher number of in context examples can give stronger ICL signal and override the effect of adversarial demonstration attacks. In line with these findings, we hypothesize that a similar mechanism operates in our experiment (Figure 7), mitigating the impact of adversarial task vector perturbations. When there is uncertainty about the underlying task basis (# of demonstrations << task dimension), both self-perturbation and adversarial perturbation are more effective.
>
> [1] Anwar, Usman, et al. "Adversarial robustness of in-context learning in transformers for linear regression." *arXiv preprint arXiv:2411.05189* (2024).
>
> [2] Cheng, Chen, et al. "Exploring the robustness of in-context learning with noisy labels." *ICASSP 2025-2025 IEEE International Conference on Acoustics, Speech and Signal Processing (ICASSP)*. IEEE, 2025.

---

### Official Review · Reviewer_RjY9 · 2025-03-22

**Overall Recommendation:** 4

**Summary:**

This paper studies how transformers learn task vectors for various tasks during pretraining. Task vectors refer to the intermediate representations from a middle layer of a transformer network, given an in-context learning (ICL) task. The paper investigates how these task encodings separate and cluster based on the task, and proposes that this separation can serve as a metric for encoding quality.

To measure this, the authors introduce Task Decodability (TD) scores, which quantify how separable and distinct the task encodings are. They show that TD scores are tightly correlated with ICL accuracy, and validate this relationship across multiple language models (e.g., Gemma, LLaMA) and tasks, including a synthetic mixture of sparse linear regression, part-of-speech tagging, and bitwise arithmetic. They also examine how task vector emergence progresses throughout pretraining using checkpointed models from OLMo-7B.

Framing ICL through a Bayesian lens, where the model learns a latent task parameter $z$, the authors argue that transformers first encode the input into a task vector in earlier layers, and then decode it to solve the task in later layers. To go beyond correlation, they apply activation patching, showing that injecting high-quality encodings improves final task accuracy, while poor encodings degrade it—further supporting the encoder-decoder interpretation of task processing in transformers. This is also bolstered by the fact that finetuning earlier layers helped task accuracies more, though this is somewhat contrary to conventional practice.

**Claims And Evidence:**

The claims presented in the paper are generally well-supported. However, there are some comments that, if addressed, could further strengthen the claims. Notably, the paper selects intermediate representations from certain layers (typically mid-range within the overall transformer architecture) but it does not articulate a clear, prescriptive criterion for choosing these layers. Establishing such a criterion for layer selection in relation to task encoding would enhance the falsifiability and robustness of the paper’s hypotheses. As it stands, the implication is that any layer compatible with the encoder-decoder framework may suffice, which weakens the specificity and testability of the claims. Or should I assume the most representative TD score is the maximum TD score over all layers, as shown in Figure 16?

**Essential References Not Discussed:**

Not that I know of.

**Experimental Designs Or Analyses:**

The experiments seem sound.

**Methods And Evaluation Criteria:**

Yes, the proposed methods and evaluation criteria are generally well-aligned with the problem at hand. The paper begins with synthetic in-context learning (ICL) experiments, building on standard regression-based ICL setups from prior work. By partitioning the base tasks and constructing a mixture of sparse linear regression problems, the authors effectively explore how task representations are encoded and separated. The naturalistic experiments are also reasonable and conceptually consistent with the research goals, though they tend to be relatively simple (bordering on toy examples), especially when contrasted with more complex, real-world benchmarks like MMLU. I would definitely like to see the same experiments interrogated on tasks like MMLU per subject.

**Other Comments Or Suggestions:**

None. See below for questions.

EDIT: Score adjusted after rebuttal (as of April 8th).

**Other Strengths And Weaknesses:**

**Strength:**
- Novel Quantitative Metric (Task Decodability): The paper introduces Task Decodability (TD) as a compelling new measure of how well intermediate representations capture task-specific structure. This metric is shown to correlate strongly with ICL performance, adding a valuable tool to interpret how transformers process in-context learning tasks.
- Encoder-Decoder Interpretation of ICL: Through activation patching and partial fine-tuning, the authors provide convincing evidence that transformers operate in an “encoder-decoder” manner—where early layers encode the task into a latent vector, and later layers decode it to generate predictions. This mechanistic perspective adds clarity to how ICL unfolds.
- Consistency Across Models and Tasks: The authors validate their findings on multiple transformer architectures (e.g., Gemma, LLaMA) and both synthetic and naturalistic tasks, which suggests a degree of generality and strengthens confidence in the interpretability of TD scores.

**Weakness**
- Investigation only on Transformer: While Transformers are the de facto model, synthetic ICL experiments have been conducted for various other architectures like LSTM and Mamba. The encoder-decoder framework needs further investigation on other architectures. Explained more in "Questions For Authors".
- Relatively Simple Task Suite: The naturalistic and synthetic tasks used (e.g., sparse linear regression, part-of-speech tagging) are conceptually valid but still fairly simple. Testing TD on more complex tasks like MMLU or richer multi-task settings could better showcase the scope and robustness of the proposed method.
- Unclear Layer Selection Criterion: While the paper observes that middle layers often provide more “task-relevant” representations, it does not prescribe a systematic way to choose these layers. A more principled method for identifying the “best” layer (or set of layers) would strengthen the paper’s claims.
- Novelty: I am not entirely familiar with the mechanistic interpretability of ICL works, but what differentiates this paper from previous works that also examine loss plateaus and representations? e.g., https://arxiv.org/pdf/2412.01003 off the top of my head. A more detailed related work comparing and contrasting their work to prior work would help the reader situate this work.

**Questions For Authors:**

I have listed some concerns and questions previously. I will only add some questions that I have not asked yet. In order of decreasing importance.

1. **Other Architectures.**
It doesn't seem that this paper performs any unique analysis for transformers and this means that their method can be applied to other architectures as well. We can extract intermediate representations of simple architectures like LSTMs or even more modern ones like Mamba in similar fashion. Does this paper's hypotheses apply to other architectures and can the experiments yield any insight into the fundamental ICL capabilities of other models?
In the case that TD scores do or do not correlate well with ICL accuracies for these architectures, I think there's something interesting to be said or investigated here.

2. **Synthetic ICL experiments.**
I find the construction to the synthetic ICL experiment mathematically appealing. But empirically, I would also like to see training a network on a mixture of different tasks, rather than just a mixture of sparse linear regressions. For example, Kim et al. (https://arxiv.org/pdf/2410.05448) train multiple tasks together (linear+quadratic+ReLU regression) and I would like to see how the encoding separations evolve here. This would strengthen the results to more diverse tasks.

**Relation To Broader Scientific Literature:**

This work introduces Task Decodability (TD) as a quantitative measure of how well task-specific information is captured in intermediate representations, linking it tightly to ICL performance. This complements mechanistic interpretabiliy studies on ICL by providing concrete evidence that task-relevant encodings emerge early in the network and are critical for downstream accuracy. The paper’s encoder-decoder framing of task processing echoes Bayesian interpretations of ICL and deepens our understanding of how transformers internally represent and differentiate between tasks during pretraining.

**Theoretical Claims:**

N/A. The only issue I have here is the notation and definitions presented in the supplementary material for Section B. First, TD score is not formally defined in the main body as far as I can tell; plus, the connection to Section B is not clear as it does not define "score" there either. $D_f$ is ill-defined and probably represents some distribution metric, though $f$ is also unclear.

---

> ### Author Rebuttal · Authors · 2025-03-31
>
> Thank you for your positive and thoughtful review. We appreciate your recognition of Task Decodability as “a compelling new measure” and that our activation patching analysis “provides convincing evidence [of transformers’] encoder-decoder structure.” We also value your suggestion to generalize to Mamba, and address your comments and questions below.
>
> > What is the criterion for layer selection for task vector analysis?
> >
>
> **Common Point #1 Raised by Reviewer bnoD,  RjY9, UVog**
>
> We select the layer with the highest TD score as the best encoding of a task. For the synthetic task (Section 3.3), we computed TD scores across all layers (Figure 10) and chose the fifth layer (layer index 4 in zero-based Figure 11). UMAP visualizations in Figure 11 confirm that representations become separable precisely when the TD score peaks.
>
> For clarity, figures illustrating TD scores and UMAP visualizations are:
>
> - Synthetic tasks : Figure 10, Figure 11
> - Natural language tasks : Figure 16
>
> > Does the work generalize to various other architectures like Mamba?
> >
>
> We **explored Mamba-2 8B and found our insights to generalize**. We conducted the same studies on natural tasks on the pretrained Mamba-2 8B model (results in https://sites.google.com/view/icl-encoder-decoder/home). The results show that the TD score correlates with ICL accuracy (Figure A), and separable representations emerges around layer 33 (Figure B). These results suggest that our encoder–decoder framework and the predictive power of TD score on ICL performance do indeed generalize beyond standard transformer-based models to recurrent models.
>
> > Selected natural tasks are simple, so can you try on MMLU per subject?
> >
>
> We refer the reviewer to **Common Point #2** in rebuttal for reviewer EJ7T. In summary, in benchmark like MMLU the boundaries between tasks are inherently blurred, making it challenging to rigorously study the encoder-decoder framework compared to our controlled experiments.
>
> > What differentiates this work from previous studies on mechanistic interpretability of ICL, such as Core et al. (2024)?
> >
>
> We concisely compare our contributions from prior work:
>
> 1. **Different Task Setup**
>     - Core et al. (2024): Adopts a relatively homogeneous Markov chain task and examines how the system transitions between different algorithmic stages (unigram/bigram retrieval and inference).
>     - **Our Work**: Focuses on the *heterogeneous tasks*—a setting more reflective of how LLMs are trained and used in practice (where tasks vary substantially). We analyze emergence of distinct task vectors during pretraining.
> 2. **Measured quality of task-specific vectors and its relation to downstream ICL peformance**
>     - Hendel et al. & Todd et al.: Demonstrate that large language models have *task vectors* but leave open questions about how these vectors emerge and why their effectiveness varies across tasks.
>     - **Our Work**: Addresses these open questions by analyzing *when and how* “task-specific” representations emerge, and how their “quality”(TD score) can be measured and used to predict downstream ICL performance.
>
> > Try Kim et al. experimental setup with different regression tasks
> >
>
> As per the reviewer’s suggestion, we replicated the regression task mixture experiment from Kim et al., training a small transformer on linear, quadratic, sparse linear, and leaky ReLU regression tasks (results available at https://sites.google.com/view/icl-encoder-decoder/home). The model successfully learns the different tasks but fails quadratic regression (Figures A, C). TD scores and UMAP visualizations (Figures B, D, E) indicate no clear separation of representations, suggesting the absence of distinct task vectors. We interpret these findings as evidence that the transformer employs similar underlying algorithms across these tasks (e.g., implementing leaky ReLU via linear regression with sign adjustments). This is consistent with Kim et al.’s observation that a common structure between their regression tasks facilitates learning (Figure 6 and Section 4.2 in their paper).
>
> > Can you clarify the notation and definition in supplementary material for Section B?
> >
>
> We will improve the decodability definition like the following in Appendix B.
>
> B.3 (Decodability). For a given decoder $G: \mathbb{R}^{d_{emb}} \rightarrow \mathcal{Z}$ and a specific latent variable $z$, the decodability measures how accurately the correct latent variable is inferred from representations. Representation is distributed as $E(z,D)$, and the inferred latent variable is  $\hat{z} \equiv G(E(z,D))$ .
>
> 1. One-hot Accuracy : $A_{\text{1-hot}}(z) = E[\mathbf{1}\bigl[\hat{z} = z\bigr]]$
> 2. f-divergence : $A_{f}(z) = D_{f}(\hat{z} \parallel z)$, where $f$ is some f-divergence metric
>
> Our Task Decodability is the one-hot accuracy with decoder of k-nearest neighbor majority-voting algorithm.
>
> For  a definition of Task Decodability, we refer reviewer to **Common Point #3** in rebuttal to Reviewer EJ7T.

---

> > ### Comment · Reviewer_RjY9 · 2025-04-02
> >
> > Thank you very much for the detailed rebuttal. I appreciate all the details and the additional experiments and plots.
> >
> > **Acknowledged:**
> > - The mamba experiments are super interesting and glad it worked out.
> > - The definitions were vague, which is why multiple reviewers asked about them. Updating the manuscript in accordance would be very helpful (upon acceptance).
> > - Thank you for the detailed comparison with related work.
> >
> > **Questions:**
> >
> > I am willing to increase my evaluation to "4: Accept" after a response to these questions.
> >
> > >  in benchmark like MMLU the boundaries between tasks are inherently blurred, making it challenging to rigorously study the encoder-decoder framework
> >
> > While this is true, not being able to study more practical benchmarks/tasks suggests that the authors' proposed framework has an inherent limitation. There are many ways to separate tasks or skills from a benchmark dataset. For example, one can separate math problems into different skills, which would correspond to different algorithms or thinking processes (https://arxiv.org/abs/2407.21009). I would find it highly motivating to apply the encoder-decoder framework to more complex cases. On the other hand, I don't believe the lack of this suggests any detriments, but rather including them would make it stronger. Just to be clear, I do not expect experiments of these kinds for my score re-evaluation.
> >
> > > We interpret these findings as evidence that the transformer employs similar underlying algorithms across these tasks (e.g., implementing leaky ReLU via linear regression with sign adjustments)
> >
> > First, thank you for running these experiments. I appreciate it very much. However, either these experiments don't make sense to me or goes against the main findings of this paper. Let's take aside the quadratic regression task since the model does not perform well on that.
> > 1) Shouldn't the encoder-decoder phenomenon still emerge even if the transformer uses the same underlying algorithm across tasks? For the mixture of sparse linear regression, transformer should be using the same essential algorithm to each subspace, yet we see separation in task vectors.
> > 2) While the trained transformer seems to high ICL accuracy for three tasks (linear, sparse linear, leaky ReLU), the TD score is low and does not occur in the middle layers.
> >
> > Thank you.

---

> > > ### Author Response · Authors · 2025-04-05
> > >
> > > Thank you for engaging in such a fruitful discussion. We try to address the reviewer’s follow-up questions in the following.
> > >
> > > > I would find it highly motivating to apply the encoder-decoder framework to more complex cases. On the other hand, I don't believe the lack of this suggests any detriments, but rather including them would make it stronger. Just to be clear, I do not expect experiments of these kinds for my score re-evaluation.
> > > >
> > >
> > > We agree that it would be very interesting to analyze how pretrained models represent different primitives in domains like mathematics. Such an investigation could offer a deeper understanding of the encoder-decoder framework’s ability to decompose and recombine reasoning components. The primary limitation, as we noted, stems from the fact that standard benchmarks such as MMLU and GSM8k do not come with explicit labels or annotations for these sub-skills, making it non-trivial to isolate and study them systematically. Constructing such labeled subsets would require significant additional effort in filtering or manual annotation. The Math Square dataset (Shah et al.), which contains curated labels of the sub-skills necessary for each problem, seems like a fitting dataset to explore. However, its small dataset size (N=210) compared to the number of sub-skills (114 sub-skills) may limit our ability to robustly quantify the representation separability per sub-skills. **We will explore this direction to extend our framework to more complex domains in future work, and appreciate the reviewer’s suggestion.**
> > >
> > > > Shouldn't the encoder-decoder phenomenon still emerge even if the transformer uses the same underlying algorithm across tasks? The TD score is low and does not occur in the middle layers.
> > >
> > > To clarify what we mean by algorithms, we adopt the definition from Li et al. [1] and refer to to the function internally implemented by the transformer, not the theoretical algorithm that can solve the problem.
> > >
> > > As the reviewer pointed out, our key observation is that task vectors do *not* emerge naturally in this mixture of regression task experiment. Our conjecture, in line with arguments from Kim et al. in Section 4.2.3, is that there is substantial structural overlap between tasks like linear regression, sparse linear regression, and leaky ReLU. All of these can be solved by implementing a linear regression algorithm [1] —sparsity can be ignored, and leaky ReLU can be approximated via simple sign adjustments—suggesting that the transformer may not need to develop distinct mechanisms for each. This may also explain the model’s failure to solve quadratic regression, which fundamentally requires a different class of function.
> > >
> > > **To further substantiate this conjecture, we conducted attention head pruning experiments and published the results on the anonymous website**: https://sites.google.com/view/icl-encoder-decoder/home (Figures I and J). We pruned each attention head sequentially and measured the resulting change in mean squared error (MSE) for each basis in synthetic sparse linear regression tasks:
> > >
> > > - **Sparse Linear Regression Tasks (Figure I):** At the fifth layer (labeled as "layer 4" in the figure, since indexing begins from zero), different attention heads distinctly corresponded to different bases. For instance, head 5 of layer 4 (l4h5) corresponds specifically to base 0, whereas head 3 of the same layer (l4h3) corresponds to base 2. We argue that these experiments provide **direct evidence that structurally distinct four-base linear regression algorithms are implemented within the model for sparse linear regression tasks**.
> > > - **Mixture of Regression Families (Figure J):** Notably, except for the quadratic regression task (which the model failed to solve effectively), we observed consistent sharing of attention heads across linear regression, leaky ReLU, and sparse linear regression tasks. This suggests structural similarity in the algorithms underlying these three regression variants, aligning closely with Kim et al.'s observation of a "common structure." We argue that these experiments provide **direct evidence that these three variants of linear regression are structurally indistinguishable within the model**.
> > >
> > > That said, the broader take-away is the following: **We believe the question of why some synthetic environments lead to the emergence of task vectors is orthogonal to our research question, albeit an interesting one. It seems that real-world language data does lead to the emergence of task vectors (Todd et al., Hendel et al.), and, in this work, we focus on how these task vectors manifest themselves during pretraining (OLMo checkpoints in Section 4.3) and how their varying quality can be measured by TD and predictive of downstream ICL performance (Section 4.2).**
> > >
> > > [1] Li, Yingcong, et al. "Transformers as algorithms: Generalization and stability in in-context learning." *International conference on machine learning*. PMLR, 2023.

---

### Decision · Program_Chairs · 2025-05-01

**Decision:**

Accept (spotlight poster)

**Comment:**

This paper investigates how transformers develop task vector representations during pretraining and their impact on in-context learning (ICL) performance. The authors propose an encoder-decoder framework (not to be confused with the encoder-decoder Transformer architecture. The paper still focuses on decoder-only Transformer architecture.) where earlier layers encode inputs into task vectors while later layers decode these representations to solve specific tasks. They use kNN classification accuracy as Task Decodability (TD) scores to quantify the separability of task encodings and demonstrate a strong correlation between TD scores and ICL accuracy across multiple language models and tasks. Through careful experiments with synthetic tasks and real LLMs, they validate this relationship and show that finetuning earlier layers improves task accuracy.

The reviewers unanimously recommend acceptance based on several strengths. The paper provides a compelling mechanistic interpretation of ICL through the encoder-decoder lens, supported by robust evidence from activation patching and partial fine-tuning experiments. The consistency of findings across different model architectures and both synthetic and naturalistic tasks demonstrates generality. The systematic investigation of task vector formation over the course of pretraining through OLMo-7B checkpoints was also highlighted as particularly insightful, providing concrete evidence that task-relevant encodings emerge early in the network and are critical for downstream performance.